# Improved Discretization Complexity Analysis of Consistency Models: Variance Exploding Forward Process and Decay Discretization Scheme

Ruofeng Yang [1]  Bo Jiang [1]  Cheng Chen [2]  Shuai Li [1]

## Abstract

Consistency models, a new class of one-step generative models, have shown competitive performance with multi-step diffusion models. The most challenging part of consistency models is the training process, which discretizes the continuous diffusion process into $K$ steps and trains a one-step mapping function on these discretized timepoints. Despite the empirical success, only a few works focus on the discretization complexity $K$, and their setting is far from that of empirical works. More specifically, the current theoretical works analyze the variance preserving (VP) diffusion process with a uniform stepsize, while empirical works adopt a variance exploding (VE) process with a decay discretization stepsize. As a result, these works suffer from large discretization complexity and fail to explain the empirical success of consistency models. To close the gap between theory and application, we analyze consistency models with (1) VE process and (2) decay stepsize and prove the state-of-the-art discretization complexity for consistency models. This result is competitive with the results of diffusion models and shows the potential of consistency models. To balance the computation and performance, previous empirical work further proposes a 2-step consistency algorithm. In this work, we also analyze the role of 2-step sampling and show that it improves the discretization complexity compared with one-step generation.

## 1. Introduction

Recently, diffusion models have shown impressive performance in different areas such as image generation and video generation (Rombach et al., 2022; Esser et al., 2024; Ho et al., 2022; Ma et al., 2024; Chen et al., 2024). The mathematical mechanism of diffusion models is made up of two processes: the forward and reverse process. The forward process gradually injects noise into data till the marginal distribution is close to pure noise. The reverse process is an iterative sampling process, which sequentially removes noise from data to generate samples. At each denoised step, diffusion models only need to predict and remove a small noise, making the training process more stable than Generative Adversarial Networks (GAN) (Goodfellow et al., 2014). However, the iterative sampling process requires diffusion models to evaluate a large neural network to predict noise at each step, leading to a higher computational cost than other one-step algorithms such as GAN, Variational Auto-Encoder (Kingma and Welling, 2013), and Normalizing Flow (Papamakarios et al., 2021).

To solve the computational issue, a series of works try to accelerate the sampling process of diffusion models (Song et al., 2020a; Zheng et al., 2023). One notable algorithm in these works is Consistency Model (Song et al., 2023), which tries to find a mapping function (a.k.a. consistency function) to directly map any points at any time of the forward process to the target data distribution. Consistency models have shown state-of-the-art (SOTA) performance compared to other one-step generative models in image generation (Song et al., 2023; Kim et al., 2024; Lu and Song, 2024), video generation (Wang et al., 2023) and music generation (Fei et al., 2024). Furthermore, it is also widely used in other areas, such as reinforcement learning (Ding and Jin, 2024) and medical image segmentation (Zhang et al., 2024).

To obtain the consistency function, consistency models discretize the timeline into $K$ discretization points and hope the consistency function outputs similar results at two adjacent points in the training phase. When the two adjacent points are far away, the learned consistency function is far away from the ground truth (Thm. 1, (Song et al., 2023)). However, it is also necessary to avoid a large $K$ since it will make the training phase time-consuming. Hence, one core of consistency models is the discretization number $K$ in the training phase, which is helpful in effectively training a consistency function with great performance.

[1]John Hopcroft Center for Computer Science, Shanghai Jiao Tong University [2]East China Normal University. Correspondence to: Shuai Li <shuaili8@sjtu.edu.cn>.

*Proceedings of the 42$^{st}$ International Conference on Machine Learning*, Vancouver, Canada. PMLR 267, 2025. Copyright 2025 by the author(s).

Despite the empirical success of consistency models, no existing works explain why consistency models achieve comparable performance to diffusion models. Though some impressive works analyze the discretization complexity of consistency models (Lyu et al., 2024; Li et al., 2024a; Dou et al., 2024), the setting is far away from the consistency models with great performance in application:

- **The forward process.** Previous theoretical works choose variance preserving forward process (VPSDE). On the contrary, empirical consistency models adopt the variance exploding forward process (VESDE) with a specific noise schedule whose solution trajectory is linear. As discussed in Karras et al. (2022) and Liu et al. (2023a), a linear trajectory is important for a one-step model (Details in Sec.D.1).

- **The discretization scheme.** The discretization scheme of consistency models is EDM (Eq.5), which first uses a large stepsize and gradually reduces the stepsize. However, Lyu et al. (2024) and Dou et al. (2024) use a uniform stepsize, and Li et al. (2024a) use a scheme that relies heavily on VPSDE (Details in Sec. D.2).

Due to the mismatch between the theoretical and empirical setting, the current discretization complexity results are significantly worse than diffusion models (Table 1). Hence, the following natural problem remains open:

*Under a setting closer to empirical works, is it possible to achieve a discretization complexity comparable to diffusion models and explain the success of consistency models?*

In this work, we answer this question by analyzing consistency models with VE forward process and EDM stepsize.

**Theorem 1.** *With mild assumptions on pre-trained score function, consistency function, consistency models require[1]*

$$K = O\left(L_f / \epsilon_{W_2}^{3 + \frac{2}{a}}\right)$$

*discretization steps in the training phase to output a distribution which is Wasserstein-2 ($W_2$) close to the target distribution in the sampling phase, where $L_f$ is Lipschitz constant of consistency function and $a$ is EDM parameter.*

As shown in Table 1, this result is better than the previous theoretical guarantee of consistency models, shows the benefit of a suitable $a$, and achieves competitive results compared to diffusion model results (Remark 4.9). The core step of this result is to make full use of the time-dependent score perturbation lemma instead of the previous uniform one. More specifically, we first show that with a uniform one, we achieve $L_f / \epsilon_{W_2}^9$ discretization complexity for any EDM parameter $a$ and does not match the empirical observation

---

[1]Here, we ignore data diameter $R$ and dimension $d$ for clarity.

(Karras et al., 2022). On the contrary, with a time-dependent score perturbation lemma, we achieve SOTA complexity $L_f / \epsilon_{W_2}^{3 + 2/a}$ for consistency models, where the influence of $a$ is highlighted (More details in Sec. 4.4).

To improve the sampling quality of consistency models, Song et al. (2023) further provide a 2-step sampling algorithm, which has been widely used in current works (Song and Dhariwal, 2023; Lu and Song, 2024). The algorithm first adds noise to clean samples generated by consistency models and obtains the noised samples. Then, the algorithm again maps the noised sample to the clean samples (Eq. 6). The great performance of this method indicates that the requirement of $K$ can be relaxed if the 2-step sampling method is used. Recently, Lyu et al. (2024) show that the 2-step sampling algorithm can reduce the $W_2$ error. However, they do not obtain an improved $K$, which does not match the empirical observation. In this work, we show that with a VESDE process, the 2-step sampling algorithm can effectively reduce $K$ to $O(L_f / \epsilon_{W_2}^{3 + 3/2a})$, which is better than Thm 4.7 and explains the role of 2-step sampling.

In conclusion, we explain why consistency models have competitive performance compared to diffusion models from the theoretical perspective. More specifically,

- We close the gap between theory and application by analyzing consistency models with the VESDE forward process and EDM discretization scheme.

- Under the above setting, we achieve the SOTA discretization complexity for consistency models, which is also competitive with the results of diffusion models.

- For the first time, we also show that the 2-step sampling algorithm can effectively reduce the discretization complexity of consistency models.

## 2. Related Work

Since the mathematical mechanism of consistency models is close to diffusion models, we discuss the discretization complexity of diffusion and consistency models. For diffusion models, we summarize the results for reverse PFODE due to the deterministic sample process of consistency models.

Before the discussion, we first discuss the log-concave assumption, which is much stronger than our bounded support assumption since it precludes the existence of multi-modal real-world data. Furthermore, under log-concave distribution, $\nabla \log q_{T-t}(\cdot)$ dose not go to $+\infty$ at the end of the reverse process, which does not match the blow-up phenomenon of score function in application (Kim et al., 2021). As a result, Gao and Zhu (2025) ignore the influence of early stopping parameter $\delta$ and an additional Poly($\epsilon_{W_2}$) (Table 1).

| Model | Forward Process | Stepsize | Complexity | Reference |
|---|---|---|---|---|
| Diffusion | VESDE | Uniform | $1/\epsilon_{W_2}^4$ 
 $1/(\epsilon_{W_2}^4\mathrm{Poly}(\epsilon_{W_2}))$ (+) | Gao and Zhu (2025) |
| | VPSDE | Uniform | $1/\epsilon_{W_2}$ 
 $1/(\epsilon_{W_2}\mathrm{Poly}(\epsilon_{W_2}))$ (+) | |
| | VPSDE (Reverse SDE) | Exponential Decay | $1/\epsilon_{W_2}^4$ | Chen et al. (2023a) |
| Consistency | VPSDE | Uniform | $L_f/\epsilon_{W_2}^7$ | Lyu et al. (2024) |
| | | Specific to VPSDE | $L_f^3/\epsilon_{W_1}$ | Li et al. (2024a) |
| | | Uniform | $L_f^2 L_{\mathrm{score}}^2/\epsilon_{W_1}^2$ 
 $L_f^2/(\epsilon_{W_1}^{10})$ (*) | Dou et al. (2024) |
| | VESDE | EDM (5), $a \in [1, \infty)$ | $L_f/\epsilon_{W_2}^{3+\frac{2}{a}}$ | Theorem 4.7 |
| | | | $L_f/\epsilon_{W_2}^{3+\frac{3}{2a}}$ | Corollary 4.12 (2-step Sampling) |
| | | Exponential Decay | $L_f/\epsilon_{W_2}^3$ | Corollary 4.8 |

Table 1: The discretization complexity for consistency and diffusion models with reverse PFODE in Wasserstein distance. To make a thorough comparison, we also provide the SOTA discretization results for diffusion models with reverse SDE. (+) means that we transform the log-concave distribution (discussed in Section 2) to our bounded support distribution. (*) means that we transform the results with $L_{\mathrm{score}}$ into the results under our setting. We present more detail in Appendix D.2.

**Diffusion models with reverse PFODE.** With strong assumptions or additional components, a series of works achieve polynomial complexity for reverse PFODE (Li et al., 2023; Chen et al., 2023c; Gao and Zhu, 2025). More specifically, Li et al. (2023) assume an accurate enough Jacobian matrix, and Gao and Zhu (2025) assume the target data distribution is log-concave. Without any strong assumption, Chen et al. (2023c) introduce a predictor-corrector algorithm, which switches between a Langevin corrector and a PFODE predictor. Then, they prove a polynomial discretization complexity for this algorithm.

**Consistency models.** Lyu et al. (2024) and Li et al. (2024a) analyze the discretization complexity of consistency distillation and consistency training paradigm, respectively. More recently, Dou et al. (2024) analyze the estimation error and discretization complexity of consistency distillation and training paradigm at the same time. Though these works deepen the understanding of consistency models, the setting of these works is far away from consistency models in the application and suffers from large discretization complexity compared to SOTA results of diffusion models (Table 1).

## 3. Preliminaries

Since the training phase of consistency models relies heavily on the diffusion process, we first introduce diffusion models in Sec.3.1. After that, Sec.3.2 introduces how to train a consistency model with a pre-trained diffusion model.

### 3.1. Diffusion Models

Diffusion models consist of two processes: the forward process and the reverse process (Song et al., 2020b). The forward process gradually converts data distribution to pure noise. To generate samples, diffusion models reverse the forward process and run the corresponding reverse process.

**The forward process.** Let $q_0$ denote the data distribution. The general forward process is

$$\mathrm{d}X_t = f(X_t, t)\, \mathrm{d}t + g(t)\, \mathrm{d}B_t, \quad X_0 \sim q_0\,,$$

where $(B_t)_{t \geq 0}$ is a $d$-dimensional Brownian motion, $f(X_t, t)$ is a drift coefficient, and $g(t)$ is a diffusion coefficient. Let $q_t$ be the density function of $X_t$ at time $t$ and $\{\beta_t\}_{t \in [0,T]}$ be a non-negative non-decreasing sequence. When $f(X_t, t) = -\beta_t X_t$ and $g(t) = \sqrt{2\beta_t}$, the general forward process is instantiated as a widely used variance preserving forward process (VPSDE) (Ho et al., 2020). We note that though VPSDE plays an important role in developing diffusion models, the solution trajectory of VPSDE is curved instead of linear, which prevents it from becoming the basis of one-step generation models.

To obtain one-step generation models, consistency models choose variance exploding (VESDE) forward process in the training phase (Song et al., 2023; Song and Dhariwal, 2023), which has a linear solution trajectory under a specific noise schedule. Let $\{\sigma_t^2\}_{t \in [0,T]}$ be a non-decreasing sequence and $g(t) = \sqrt{\mathrm{d}\sigma_t^2/\mathrm{d}t}$. Then, VESDE is defined by:

$$\mathrm{d}X_t = g(t)\, \mathrm{d}B_t, \quad X_0 \sim q_0\,. \tag{1}$$

As shown in Karras et al. (2022), when choosing $\sigma_t^2 = t^2$, the solution trajectory of VESDE is linear, which indicates it is possible to generate samples with a single Euler step (Detail in Sec.D.1) (Song et al., 2023; Liu et al., 2023b). Hence, consistency models adopt VESDE ($\sigma_t^2 = t^2$) as the forward process, and we also choose VESDE with $\sigma_t^2 = t^2$ as the forward process to match the empirical setting.

**The reverse process.** Let $t' = T - t$ be the reverse time and $(Y_{t'})_{t' \in [0,T]} = (X_{T-t'})_{t' \in [0,T]}$. To generate samples, the model reverses the forward process (Eq. 1) and obtains the reverse process (probability flow ODE, PFODE) [2]:

$$\mathrm{d}Y_{t'} = \frac{1}{2}g(T-t')^2 \nabla \log q_{T-t'}(Y_{t'}) \, \mathrm{d}t', Y_0 \sim q_T. \quad (2)$$

Since the ground truth score function $\nabla \log q_t(\cdot)$ and $q_T$ contain the data information, we can not directly run the above PFODE to generate samples. For the reverse beginning distribution $q_T$, we choose $\mathcal{N}(0, T^2 I_d)$ to approximate it due to $\sigma_T^2 = T^2$. For $\nabla \log q_t(\cdot)$, Vincent (2011) propose the following score matching objective function to learn an approximated score function $s_\phi(X_t, t), \forall t \in [0, T]$:

$$\min_{\phi \in \Phi} \int_0^T \mathbb{E}_{X_0} \left[ \mathbb{E}_{X_t|X_0} \| \nabla \log q_t(X_t|X_0) - s_\phi(X_t, t) \|_2^2 \right] \mathrm{d}t.$$

With the approximated score function $s_\phi(\cdot)$, diffusion models discretize the reverse process and generate samples. Let $\delta = t_0 \le t_1 \le \cdots \le t_K = T$ be the discretization points in the forward time and $h_k := t_k - t_{k-1}$ be the stepsize. When considering the reverse process, we define by $t'_k = T - t_{K-k}$ and $h'_k = h_{K-k}$ the discretization points and stepsize in the reverse process, respectively. Since the score function $\nabla \log q_{T-t}$ goes to $+\infty$ at the end of the reverse process, we adopt the early stopping technique by setting $t_0 = \delta$ to avoid this issue, which is widely used in the application (Song et al., 2020b; Kim et al., 2021). Then, starting from $\bar{Y}_0 \sim \mathcal{N}(0, T^2 I_d)$, diffusion models run the following process in each interval $t \in [t'_k, t'_{k+1}], k \in [0, K-1]$ to generate samples:

$$\mathrm{d}\bar{Y}_{t'} = \frac{g(T-t')^2}{2} s_\phi\left(\bar{Y}_{t'_k}, T - t'_k\right) \mathrm{d}t', t' \in [t'_k, t'_{k+1}].$$

### 3.2. Consistency Models

This part introduces how to obtain a consistency function to directly map pure noise to the target distribution by using the above PFODE process and pre-trained score function. As a beginning, we first introduce the learning goal of consistency models. Let $\boldsymbol{v}(Y, t') = \frac{g(T-t')^2}{2} \nabla \log q_{T-t'}(Y)$ be the

exact vector field with the ground truth score function. Then, the reverse PFODE (Equation (2)) has the following form:

$$\mathrm{d}Y_{t'} = \boldsymbol{v}(Y_{t'}, t')\mathrm{d}t', Y_0 \sim q_T.$$

Let $\boldsymbol{f^v} : \mathbb{R}^d \times \mathbb{R}^+ \to \mathbb{R}^d$ be the associate backward mapping of the above PFODE. Then, we know that:

$$\boldsymbol{f^v}(Y_{t'}, t') = Y_{T-\delta} = X_\delta, \forall t' \in [0, T-\delta],$$

where $\delta$ is the early stopping parameter. The above equation is equivalent to the following conditions:

$$\begin{aligned} \boldsymbol{f^v}(Y_{t'}, t') &= \boldsymbol{f^v}(Y_{t''}, t''), \forall 0 \le t'', t' \le T - \delta, \\ \boldsymbol{f^v}(Y, T-\delta) &= Y, \forall Y \in \mathbb{R}^d. \end{aligned} \quad (3)$$

The goal of consistency models is to train a consistency function $\boldsymbol{f_\theta}$ to approximate $\boldsymbol{f^v}$ and do one-step generation. Let $F_{\boldsymbol{\theta}}(Y, t')$ be a free-form deep neural network. To satisfy the boundary condition (the second equation of Equation (3)), Song et al. (2023) parameter $\boldsymbol{f_\theta}$ with the following form:

$$\boldsymbol{f_\theta}(Y, t) = \begin{cases} Y & t' = T - \delta \\ F_{\boldsymbol{\theta}}(Y, t') & t' \in [0, T-\delta) \end{cases}.$$

There are two paradigms for consistency models: consistency distillation (CD) and consistency training (CT), where CD requires a pre-trained score function $s_\phi(Y, t')$ and CT trains independently (Song et al., 2023; Lu and Song, 2024). Since CT can not take information from a pre-trained score function, its hyperparameters need to be carefully selected to achieve great performance (Song and Dhariwal, 2023). On the contrary, the CD paradigm has a stable training process (Song et al., 2023). Hence, we analyze the CD paradigm and discuss the current results of CT paradigm in Remark 4.10.

**Consistency Distillation Paradigm.** Let $\hat{Y}_{t'_{k+1}}^\phi$ be the output by running one step PFODE from $t'_k$ to $t'_{k+1}$ with initial distribution $Y_{t'_k}$ and approximated score function $s_\phi$:

$$\hat{Y}_{t'_{k+1}}^\phi = Y_{t'_k} + \frac{(2T - t'_k - t'_{k+1})h'_k}{4} s_\phi(Y_{t'_k}, T - t'_k).$$

Motivated by Eq. 3, Song et al. (2023) propose the following consistency distillation objective function:

$$\mathcal{L}_{\mathrm{CD}}^K\left(\boldsymbol{\theta}, \boldsymbol{\theta}^-; \phi\right) :=$$

$$\mathbb{E}_{X_0}\left[\mathbb{E}_{Y_{t'_k}|X_0} \left\| \boldsymbol{f_\theta}(Y_{t'_k}, t'_k) - \boldsymbol{f_{\theta^-}}(\hat{Y}_{t'_{k+1}}^\phi, t'_{k+1}) \right\|_2^2 \right], \quad (4)$$

where $t'_k$ is the time discretization points in the reverse process and $k$ is uniformly distributed over $\{0, 1, ..., K-1\}$. Since $Y_{t'_k}$ is equal to $X_{T-t'_k}$, we calculate $Y_{t'_k}|X_0$ using the forward process $X_0 + (T - t'_k)Z$, where $Z$ is the standard Gaussian noise. To make the training process more

---

[2]We note that there are two kinds of the reverse process: (a) SDE and (b) PFODE (Song et al., 2020b). Since the consistency models adopt the PFODE process in the training phase, we only present this process for simplicity.

stable, (Song et al., 2023) introduce an additional parameter $\theta^-$, which is updated using an exponential moving average (EMA) strategy $\theta^- = \text{stopgrad} \left( \mu \theta^- + (1 - \mu) \theta \right)$, where $\mu \in [0, 1)$ is the decay rate. We also note that this objective function has already been adopted by many one-step flow-based models, like InstaFlow (Liu et al., 2023b).

Recently, Dou et al. (2024) discretize the interval $[t'_k, t'_{k+1}]$ in $M$ smaller intervals and run multi step PFODE to obtain $\hat{Y}^{\phi}_{t'_{k+1}}$. We note that though this operation makes theoretical analysis easier, it is far away from the real-world application and time-consuming. Since our work aims to explain the empirical success of consistency models in application, we exactly follow the empirical operation, which does one-step PFODE instead of multi-step PFODE.

**The stepsize of consistency model.** When training the consistency model, Song et al. (2023) and Song and Dhariwal (2023) use EDM stepsize

$$t_k = (\delta + kh)^a \text{ and } h = (T^{1/a} - \delta)/K, \quad (5)$$

with $a = 7$. As discussed in Karras et al. (2022), since VESDE has a large variance at the end of the forward process, it is more suitable for VESDE to use a large stepsize at the beginning of the reverse process instead of uniform stepsize ($a = 1$). When $a$ goes to $+\infty$, the EDM stepsize becomes a theoretically friendly exponential decay stepsize $h_k = rt_k$, where $r$ is a small coefficient corresponding to accuracy parameters $\epsilon$. We note that the exponential decay stepsize is widely used in theoretical works (Chen et al., 2023a; Benton et al., 2024). In this work, we simultaneously analyze the EDM and exponential decay steps and achieve state-of-the-art discretization complexity.

**Notation.** We denote by $W_1$ and $W_2$ the Wasserstein distance of order one and two, respectively. Note that $W_1$ guarantee is weaker than $W_2$ guarantee since $W_1(p, q) \leq W_2(p, q)$. The push-forward operator $\sharp$ is associated with a measurable map $f : \mathcal{M}' \to \mathcal{N}$. For any measure $\mu$ over $\mathcal{M}'$, we define the push-forward measure $f\sharp\mu$ over $\mathcal{N}$ by: $f\sharp\mu(A) = \mu \left( f^{-1}(A) \right)$, for any $A$ be measurable set in $\mathcal{N}$. A complete notation part is provided in Appendix A.

# 4. Discretization Complexity of Consistency Model

The goal of consistency models is to train a consistency function $\boldsymbol{f_\theta}(Y, t')$ to directly map pure noise $Y \sim \mathcal{N}(0, T^2 I_d)$ and $t' = 0$ to target distribution $q_0$. Let $\boldsymbol{f_{\theta,0}}$ be the learned consistency function at $t' = 0$ and $\boldsymbol{f_{\theta,0}}\sharp\mathcal{N}\left(0, T^2 I_d\right)$ be the generated distribution. In the training phase, consistency models discretize the continuous time $t' \in [0, T - \delta]$ into $K$ intervals and use the CD objective function to learn a consistency function (Eq.4). The goal of this work is to determine the requirement of $K$ in the train-

ing process (called discretization complexity) to guarantee $W_2(\boldsymbol{f_{\theta,0}}\sharp\mathcal{N}\left(0, T^2 I_d\right), q_0) \leq \epsilon_{W_2}$. This section provides state-of-the-art discretization complexity for one-step consistency models (Sec.4.2) and shows that the multi-step (2-step) sampling algorithm proposed by Song et al. (2023) can further reduce the discretization complexity (Sec.4.3).

## 4.1. Assumptions

Before showing our results, we introduce some suitable assumptions on data distribution, pre-trained score function, and consistency function.

**Assumption 4.1.** $q_0$ is supported on a compact set $\mathcal{M}$, where $0 \in \mathcal{M}$ and $R = \sup\{\|x - y\|_2 : x, y \in \mathcal{M}\} \geq 1$.

The bounded support assumption is naturally satisfied by the image dataset and is widely used by theoretical works (De Bortoli, 2022; Lyu et al., 2024; Yang et al., 2024).

For the approximated score, similar to previous works, we assume it is $L_2$-accurate (De Bortoli, 2022; Lyu et al., 2024).

**Assumption 4.2.** There exists a constant $\epsilon_{\text{score}}$ such that for any $k \in [K]$,

$$\mathbb{E}_{X_{t_k} \sim q_{t_k}} \left[ \|s_\phi(X_{t_k}, t_k) - \nabla \log q_{t_k}(X_{t_k})\|_2^2 \right] \leq \epsilon_{\text{score}}^2 / \sigma_{t_k}^2.$$

We also assume after the one-step reverse PFODE process, the output of the learned consistency function are still close.

**Assumption 4.3.** There exists a constant $\epsilon_{\text{cm}}$ such that for any $k \in [K]$ and $Y_{t'_k} \sim q_{t'_k}$

$$\mathbb{E}\left[ \left\| \boldsymbol{f_\theta}(Y_{t'_k}, t'_k) - \boldsymbol{f_\theta}(\hat{Y}^{\phi}_{t'_{k+1}}, t'_{k+1}) \right\|_2^2 \right] \leq \epsilon_{\text{cm}}^2 \left( t'_{k+1} - t'_k \right)^2.$$

The above assumption is also used by Lyu et al. (2024). We note that this assumption has the same form with $\mathcal{L}_{\text{CD}}^K$ and can be satisfied when $\mathcal{L}_{\text{CD}}^K$ is small enough.

Similar with previous theoretical analysis, we also assume $\boldsymbol{f_\theta}(Y, t')$ is Lipschitz.

**Assumption 4.4.** $\boldsymbol{f_\theta}(Y, t')$ is $L_f$-Lipschitz for $t' \in [0, T - \delta]$ and is $L_{f,0}$-Lipschitz with $L_{f,0} = R/T$ when $t' = 0$.

The first part is a standard one used by all previous theoretical works on consistency models (Lyu et al., 2024; Li et al., 2024a; Dou et al., 2024). We further assume the second part for VESDE forward process. In the following two paragraphs, we first provide empirical evidence to support our additional assumption (Example 4.5). Then, we discuss why the additional assumption is necessary for consistency models with VESDE (Remark 4.6).

*Example* 4.5. As a start, we first provide $\boldsymbol{f^v}$ for a 1-$d$ Gaussian target distribution $q_0 = \mathcal{N}(\mu, \sigma^2)$:

$$\boldsymbol{f^v}(Y_0, 0) = \mu + \frac{Y_0 - \mu}{\sqrt{\sigma^2 + T^2}} \sigma,$$

where $Y_0 \sim q_T = \mathcal{N}(\mu, \sigma^2 + T^2)$. It is clear that the Lipschitz constant $L_{f,0}$ of $\boldsymbol{f^v}$ is $\sigma/\sqrt{\sigma^2 + T^2} \le R/T$ at $t' = 0$ [3]. We note that this result also holds for high-dimension multivariate Gaussian, as shown in Li et al. (2024b).

For the highly multi-modal target distribution, the consistency function for these distributions does not have a closed form. There are two choices to overcome this hardness. The first choice is to run experiments to simulate the solution and calculate the order of Lipschitz constant. In Appendix F, we run simulation experiments on the multi-modal Gaussian mixture distribution (3-modal and 4-modal GMM) and verify the Lipschitz constant $L_{f,0}$ of $\boldsymbol{f^v}$ has an order of $1/T$. The second choice is to add some assumptions on the multi-modal target data to simplify the PFODE process and then show the order of the Lipschitz constant. In this part, we use 2-modal GMM target distribution $X_0 \sim 1/2N(\mu, \sigma^2 I_d) + 1/2N(-\mu, \sigma^2 I_d)$ as an example to support our discussion. The score of the target distribution has the following form (Appendix A.2 of Shah et al. (2023))

$$\nabla \log q_t(X_t) = \tanh\left(\frac{\mu^\top X_t}{\sigma_t^2 + \sigma^2}\right)\frac{\mu}{\sigma_t^2 + \sigma^2} - \frac{X_t}{\sigma_t^2 + \sigma^2}.$$

Since $\boldsymbol{f}^{\text{ex}}(Y_0, 0)$ the associate backward mapping of the following PFODE (in the following part, we ignore the superscript of $t'$):

$$\mathrm{d}Y_t = \frac{(T-t)}{(T-t)^2 + \sigma^2}\left(\tanh\left(\frac{\mu^\top Y_t}{(T-t)^2 + \sigma^2}\right)\mu - Y_t\right)\mathrm{d}t,$$

we need to solve it to obtain $\boldsymbol{f}^{\text{ex}}(Y_0, 0)$. To simplify the PFODE process, we assume $\mu$ is smaller enough to guarantee $\tanh\left(\frac{\mu^\top Y_t}{(T-t)^2 + \sigma^2}\right)$ can be approximated by $\frac{\mu^\top Y_t}{(T-t)^2 + \sigma^2}$, which simplify PFODE to a linear ODE (in fact, the distribution gradually closes to Gaussian)

$$\mathrm{d}Y_t = \left(\frac{\mu^\top \mu Y_t(T-t)}{((T-t)^2 + \sigma^2)^2} - \frac{Y_t(T-t)}{(T-t)^2 + \sigma^2}\right)\mathrm{d}t,$$

which have the following solution

$$Y_t = Y_0\left(\sqrt{\frac{\sigma^2 + (T-t)^2}{\sigma^2 + T^2}} \times \right.$$
$$\left. \exp\left(\frac{\mu^2}{2}\left(\frac{1}{\sigma^2 + (T-t)^2} - \frac{1}{\sigma^2 + T^2}\right)\right)\right).$$

and indicate

$$Y_T = Y_0\left(\sqrt{\frac{\sigma^2}{\sigma^2 + T^2}}\exp\left(\frac{\mu^2}{2}\left(\frac{1}{\sigma^2} - \frac{1}{\sigma^2 + T^2}\right)\right)\right).$$

---

[3] Since Gaussian is log-concave, the score does not blow-up (Details in Sec.2), and we do not use early stopping technique here.

Taking the derivative of $Y_0$, we know that the $L_{f,0}$ have order $1/T$. The intuition of $L_{f,0} = O(1/T)$ is that the variance of $\mathcal{N}(0, T^2 I)$ is much larger than data variance. Hence, one necessary step to map pure noise to the target distribution is to remove large variance by multiplying a $1/T$, which leads to an $O(1/T)$ Lipschitz constant.

*Remark* 4.6 (The necessity of $L_{f,0} = R/T$). As shown in Eq. 7, one error term is $W_2\left(\boldsymbol{f_{\theta,0}}\sharp\mathcal{N}\left(0, \sigma_T^2 I_d\right), \boldsymbol{f_{\theta,0}}\sharp q_T\right)$, which corresponds to the forward process and is bounded by $L_{f,0}W_2(\mathcal{N}(0, \sigma_T^2 I_d), q_T)$. For VPSDE forward process, as shown in Lyu et al. (2024), $W_2(\mathcal{N}(0, I_d), q_T) \le (\sqrt{d} + R)\exp(-T)$ since VPSDE converges to $\mathcal{N}(0, I_d)$ with an exponential convergence rate. On the contrary, for VESDE, we know that $W_2\left(\mathcal{N}\left(0, T^2 I_d\right), q_T\right)$ is bounded by

$$\mathbb{E}_{q_0}\left[\|X_0 + (T-T)\xi\|_2^2\right] \le R^2,$$

where $\xi \sim \mathcal{N}(0, I_d)$. It is clear that this term is independent of $T$. As a result, we need $L_{f,0}$ depends on $T$ and show that $L_{f,0}$ has order of $1/T$ (Example 4.5).

We also note that even with Assumption 4.4, the polynomial $T = 1/\epsilon_{W_2}$ for VESDE is much larger than the logarithmic $T = \log(1/\epsilon_{W_2})$ for VPSDE. However, in Remark 4.15, we show that VESDE has better order on early stopping $\delta$, which offsets the influence of large $T$. Furthermore, as shown in Thm.4.7, the dependence of $T$ is $T^{1/a}$, which indicates EDM stepsize further reduces the influence of $T$.

### 4.2. Improved Results for Consistency Models

With these assumptions, we obtain the following results under the VESDE and EDM stepsize.

**Theorem 4.7.** *Assume Assumption 4.1, 4.2, 4.3, 4.4 holds and consider the EDM stepsize (5). Then, one-step generation error $W_2\left(\boldsymbol{f_{\theta,0}}\sharp\mathcal{N}\left(0, T^2 I_d\right), q_0\right)$ is bounded by*

$$\frac{R^2}{T} + \frac{L_f R^2(R + \sqrt{d})(T/\delta)^{\frac{1}{a}}}{K\delta^2} + L_f T\epsilon_{\text{score}} + T\epsilon_{\text{cm}} + \sqrt{d}\delta.$$

*Furthermore, by choosing $T \ge R^2/\epsilon_{W_2}$, $\delta = \epsilon_{W_2}/\sqrt{d}$, $\epsilon_{\text{cm}} \le \epsilon_{W_2}/T$ and $\epsilon_{\text{score}} \le \epsilon_{W_2}^2/(L_f R^2)$, the output is $\epsilon_{W_2}$-close to $q_0$ with discretization complexity*

$$K = O\left(\frac{L_f R^{2+\frac{2}{a}}(R + \sqrt{d})d^{1+\frac{1}{2a}}}{\epsilon_{W_2}^{3+\frac{2}{a}}}\right).$$

Song et al. (2023) choose $a = 7$ for the EDM scheme, which leads to $O(L_f/\epsilon_{W_2}^{23/7})$ result. When considering exponential decay stepsize, we improve the results as follows.

**Corollary 4.8.** *Assume Assumption 4.1, 4.2, 4.3, 4.4 holds and consider the exponential decay stepsize $h_k = rt_k$, where $r = \epsilon_{W_2}^3/\left(R^2 \log^2(T/\delta)\right)$. Following $T, \delta, \epsilon_{\text{cm}}$ in*

*Thm. 4.7 and choosing $\epsilon_{\text{score}} \leq \epsilon_{W_2}\delta/(R^2 \log(T/\delta))$, the output is $\epsilon_{W_2}$-close to $q_0$ with discretization complexity*

$$K = O\left(L_f R^2 d(R + \sqrt{d}) \log^2(T/\delta)/\epsilon_{W_2}^3\right).$$

As shown in Table 1, the above discretization complexity $O(L_f/\epsilon_{W_2}^3)$ for CD paradigm significantly improves the current results $O(L_f/\epsilon_{W_2}^7)$ provided by (Lyu et al., 2024) (We discuss current results for CT paradigm in Remark 4.10.). Furthermore, as shown in Remark 4.9, this result achieves competitive results with diffusion models in both reverse SDE and PFODE settings by taking full use of the time-dependent score perturbation lemma (Sec. 4.4), which shows the potential of consistency models.

*Remark* 4.9 (The Diffusion Model Results). The SOTA results for diffusion models under $W_2$ distance is $O(1/\epsilon_{W_2}^4)$. Coro. 4.8 achieve $O(L_f/\epsilon_{W_2}^3)$ discretization and will be better than $O(1/\epsilon_{W_2}^4)$ if $L_f \leq O(1/\epsilon_{W_2})$. As shown in Thm. 4.7, the early stopping parameter $\delta$ has order $\epsilon_{W_2}$. Hence, we require $L_f \leq 1/\delta$ for $\forall t' \in [0, T-\delta]$, which can be satisfied if $L_f$ grows slower than $1/(T-t')$. As shown in Example 4.5, the Gaussian distribution satisfied this growth rate. For multi-modal real-world distributions, as shown in Appendix F, the growth rate is much slower than $1/(T-t')$, which also satisfies our requirement.

*Remark* 4.10 (The CT Paradigm Results). Li et al. (2024a) and Dou et al. (2024) analyze the discretization complexity of CT paradigm. As shown in Table 1, their discretization complexity results are worse than our results. More specifically, Li et al. (2024a) achieve a $L_f^3/\epsilon_{W_1}$ result, which adopt a weaker $W_1$ guarantee and has a worse $L_f$ dependence. Dou et al. (2024) also adopt the $W_1$ guarantee and achieve $L_f^2/\epsilon_{W_1}^{10}$ results. More importantly, their paradigm is different from the CT paradigm in application. Li et al. (2024a) use an iterative consistency training method, which trains a consistency function for each $k \in [K]$ and is time-consuming. To avoid this problem, Dou et al. (2024) only train a consistency function. However, they run multi-step PFODE in the training phase, which does not match the operation (only run one-step PFODE) of the empirical consistency models (Song et al., 2023; Song and Dhariwal, 2023). Hence, it is an interesting future work to explain the empirical success of CT paradigm from the theoretical perspective in the setting used by empirical works.

*Remark* 4.11. Recently, some works (Lyu et al., 2024; Li et al., 2024a; Dou et al., 2024) assume the second moment of data distribution $\mathbb{E}[\|q_0\|_2^2]$ is bounded, which is slightly weaker than Assumption 4.1. The dependence of $R$ comes from our time-dependent score perturbation lemma, which is an important part of making full use of EDM stepsize (Sec. 4.4). We also note that to achieve a refined analysis, Lemma 3.13 of Lyu et al. (2024) and Lemma D.2 of Dou et al. (2024) also assume Assumption 4.1 to obtain the Lipschitz constant $R^2/\sigma_{T-t'}^4$ for the score function.

### 4.3. 2-step Sampling Further Reduce Complexity

To achieve better performance, a multi-step sampling method is used by consistency models (Song et al., 2023; Lu and Song, 2024). Let $T = \tau_1 \geq \tau_2 \geq ... \geq \tau_N \geq \delta$ be a sequence of time points, $p_1$ be the one-step generated distribution $\boldsymbol{f}_{\boldsymbol{\theta},0}\sharp\mathcal{N}\left(0, T^2 I_d\right)$ and $X^{\tau_1} \sim p_1$. The $n$-step sampling process has the following procedure:

$$X^{\tau_n} = \boldsymbol{f}_{\boldsymbol{\theta}}(X^{\tau_{n-1}} + \sigma_{\tau_n}Z, \tau_n), Z \sim \mathcal{N}(0, I), \quad (6)$$

which first adds noise to the $(n-1)$-step sampling data using the VESDE forward process and then generates $X^{\tau_n}$. Let $p_n$ be $\text{law}(X^{\tau_n})$. Recently, Lyu et al. (2024) make an important step in understanding the multi-step sampling mechanism in consistency models and prove that this operation can reduce the $W_2$ error with a suitable $N$. However, as shown in Corollary 3.14 of Lyu et al. (2024), both the discretization complexity of one-step and multi-step sampling are $\tilde{O}(L_f/\epsilon_{W_2}^7)$. This result means the multi-step sampling can not reduce the requirement of discretization, which does not match the empirical observation. Hence, it motivates us to do a more refined analysis under a realistic setting and show the role of multi-step sampling. Since $n = 2$ is enough to generate high-quality samples in application (Song et al., 2023; Lu and Song, 2024), we analyze the 2-step sampling and improve discretization complexity [4].

**Corollary 4.12.** *Assume Assumption 4.1, 4.2, 4.3, 4.4 holds and $L_{f,T/2} = \Theta(R/T)$. Choosing EDM stepsize and $\tau_2 = T/2$ (2-step sampling), $W_2(p_2, q_0)$ is bounded by*

$$\sqrt{d}\delta + R^3/(\tau_2 T) + L_f R^2(R + \sqrt{d})(T/\delta)^{\frac{1}{a}}/(K\delta^2)$$
$$+ (RT/\tau_2 + \tau_2)(L_f \epsilon_{\text{score}} + \epsilon_{\text{cm}}).$$

*By choosing $T \geq R^{1.5}/\sqrt{\epsilon_{W_2}}$, $\delta = \epsilon_{W_2}/\sqrt{d}$, $\epsilon_{\text{cm}} \leq \epsilon_{W_2}/T$ and $\epsilon_{\text{score}} \leq \epsilon_{W_2}^2/(L_f R^2)$, the output is $\epsilon_{W_2}$-close to $q_0$ with discretization complexity*

$$K = O\left(L_f R^{2+\frac{3}{2a}}(R + \sqrt{d})d^{1+\frac{1}{2a}}/\epsilon_{W_2}^{3+\frac{3}{2a}}\right).$$

The error bound comes from three sources: the early stopping term, the previous error $W_2(p_1, q_\delta)$ and the discretization error at this sampling phase. The core observation is that the multi-step sampling can reduce the requirement of $T$ due to the $R^3/(\tau_2 T)$ term, which further improves the discretization complexity.

We note that compared to Thm. 4.7, we further assume $L_{f,T/2} = \Theta(R/T)$, which is also supported by our simulation experiments. The intuition is that since $T/2$ and $T$ have the same order and are both large, the Lipschitz constant at $T/2$ also should have the same $1/T$ order with $T$.

---

[4] Our analysis can be directly extended to multi-step sampling.

*Remark* 4.13 (*N*-step Sampling). We note that our analysis can be extended to *N*-step sampling algorithm and can achieve nearly $L_f/\epsilon_{W_2}^{3+1/a}$ (which is better than Thm. 4.7 and Coro 4.12) under the EDM stepsize. We use 3-step sampling algorithm as an example ($\tau_1 = T, \tau_2 = 3T/4, \tau_3 = T/2$. Here, we still further require $L_{f,3T/4} = \Theta(R/T)$). Under this setting, the result becomes (here we ignore $\epsilon_{score}, \epsilon_{cm}, R, d$ and focus on the dominated term)

$$\delta + 1/T^3 + + L_f(T/\delta)^{\frac{1}{a}} / \left(K\delta^2\right) .$$

To guarantee the above term smaller than $\epsilon_{W_2}$, we require $\delta = \epsilon_{W_2}$ and $K \geq L_f T^{1/a}/(\delta^{2+1/a}\epsilon_{W_2})$, which is the same with the one-step and two-step sampling algorithms. However, 3-step algorithm only require $T \geq 1/\epsilon_{W_2}^{1/3}$, which is better than $1/\epsilon_{W_2}$ of 1-step and $1/\epsilon_{W_2}^{1/2}$ of 2-step. Hence, the discretization complexity for 3-step sampling algorithm is $L_f/\epsilon_{W_2}^{3+4/(3a)}$, which is better than 2-step algorithm. The above steps can be extended to $N$ steps, and the influence of $T$ decreases, and finally, $T$ does not affect the discretization complexity, leading to $L_f/\epsilon_{W_2}^{3+1/a}$ results.

## 4.4. Proof Sketch and Technique Novelty

In this section, we first provide a proof sketch for Theorem 4.7 to introduce the dependence of score perturbation lemma. Then, we highlight our technique novelty to take advantage of the time-dependent score perturbation.

**Proof Sketch.** We first decompose the target error term $W_2\left(f_{\theta,0}\sharp\mathcal{N}\left(0,T^2 I_d\right), q_0\right)$ as:

$$W_2\left(f_{\theta,0}\sharp\mathcal{N}\left(0,T^2 I_d\right), f_{\theta,0}\sharp q_T\right) + W_2\left(f_{\theta,0}\sharp q_T, q_\delta\right)$$
$$+ W_2\left(q_\delta, q_0\right) , \quad (7)$$

where the first term is due to the forward process, the second term is the discretization error, and the third term is due to the early stopping $\delta$. The first term is discussed in Remark 4.6, and the third term is smaller than $\sqrt{d}\delta$. This part focuses on the discretization term $W_2\left(f_{\theta,0}\sharp q_T, q_\delta\right)$, which is controlled by the following inequality.

$$\left(\mathbb{E}\left[\|f_\theta\left(Y_0, 0\right) - f^v\left(Y_0, 0\right)\|_2^2\right]\right)^{1/2}$$
$$= \left(\mathbb{E}\left[\| \sum_{k=0}^{K-1} (f_\theta(Y_{t'_k}, t'_k) - f_\theta(Y_{t'_{k+1}}, t'_{k+1}))\|_2^2\right]\right)^{1/2}$$
$$\leq \sum_{k=0}^{K-1} \left(\mathbb{E}\left[\|f_\theta(\hat{Y}_{t'_{k+1}}^\phi, t'_{k+1}) - f_\theta(Y_{t'_{k+1}}, t'_{k+1})\|_2^2\right]\right)^{1/2}$$
$$+ \sum_{k=0}^{K-1} \left(\mathbb{E}\left[\|f_\theta(Y_{t'_k}, t'_k) - f_\theta(\hat{Y}_{t'_{k+1}}^\phi, t'_{k+1})\|_2^2\right]\right)^{1/2}$$
$$\leq \sum_{k=0}^{K-1} L_f \left(\mathbb{E}\left[\|\hat{Y}_{t'_{k+1}}^\phi - Y_{t'_{k+1}}\|_2^2\right]\right)^{1/2} + \epsilon_{cm}T , \quad (8)$$

where the expectation is taken over $Y_0 \sim q_T$ and the first equality follows the fact that $f^v\left(Y_0, 0\right) = X_\delta = f_\theta\left(Y_{T-\delta}, T - \delta\right)$. After that, we aim to control the error bound of one step PFODE, which relies heavily on $\mathbb{E}_{X_t \sim q_t}\left[\|\partial_t\nabla \log q_t\left(X_t\right)\|_2\right]$:

$$\mathbb{E}_{Y_0 \sim q_T}\left[\|\hat{Y}_{t'_{k+1}}^\phi - Y_{t'_{k+1}}\|_2^2\right] \leq h_k'^2 \epsilon_{score}^2$$
$$+ \int_{t'_k}^{t'_k+h'_k} 4(T-t)^2 h_k'^3 \left[\|\partial_t\nabla \log q_{T-t}\left(Y_t\right)\|_2^2\right] dt . \quad (9)$$

In the following paragraph, we focus on the second term and discuss why a time-dependent lemma is suitable and necessary for EDM stepsize to achieve an improved results.

**Time-dependent Score Perturbation Lemma.** Following previous works, we call the following lemma score perturbation lemma. However, different from previous lemma uniformly holds for $t \in [\delta, T]$ (Lyu et al., 2024; Chen et al., 2023c), we provide the following time-dependent score perturbation lemma, which leads to an improved results.

**Lemma 4.14.** *For VESDE (Eq. 1), we have that*

$$\mathbb{E}_{X_t \sim q_t}\left[\|\partial_t\nabla \log q_t\left(X_t\right)\|_2\right] \leq R^3/t^5 + R^2\sqrt{d}/t^4 .$$

With this lemma, we can control the one step PFODE error (Eq. 9) and then the discretization error.

Before showing the advantage of the time-dependent lemma, we first provide the result with the uniform score perturbation lemma, which has large discretization error and does not reflect the role of EDM. By setting $t = \delta$ in Lem. 4.14, we obtain an uniform version lemma holds for $t \in [\delta, T]$:

$$\mathbb{E}_{X_t \sim q_t}\left[\|\partial_t\nabla \log q_t\left(X_t\right)\|_2\right] \leq R^2(R + \sqrt{d})/\delta^5 .$$

Combined with Eq. 8 and Eq. 9, we know that the first term of Eq. 8 is smaller than

$$\frac{L_f T R^2(R + \sqrt{d})}{\delta^5} \sum_{k=0}^{K-1} h_k^2 .$$

We know that

$$\sum_{k=0}^{K-1} h_k^2 \asymp h \sum_{k=0}^{K-1} h_k t_k^{\frac{a-1}{a}} \asymp \frac{T^{1/a}}{K} \int_\delta^T t^{\frac{a-1}{a}} \asymp \frac{T^2}{K} , \quad (10)$$

where the first equality follows the fact that $h_k/h \asymp t_k^{\frac{a-1}{a}}$ and the equality comes from $h = (T^{1/a} - \delta)/K$. To make this term smaller than $\epsilon_{W_2}$, we require $K \geq L_f T^3/(\epsilon_{W_2}\delta^5)$ [5]. With $T$ and $\delta$ in Thm. 4.7, we know that $K$ has order of $L_f/\epsilon_{W_2}^9$, which has a large dependence on $\epsilon_{W_2}$.

---

[5] Here, we only focus on $\epsilon_{W_2}$ and ignore $R$ and $d$ for clarity.

More importantly, it is clear that the result with uniform version lemma is not influenced by EDM parameter $a$. However, Karras et al. (2022) have shown that a suitable $a = 7$ can significantly improve the generation quality, and consistency models adopt this setting (Song et al., 2023). In this part, we show that with the time-dependent score perturbation lemma, the discretization error can be well control to achieve an improved complexity. More specifically, with Lem. 4.14, we have that

$$
\sum_{k=0}^{K-1} L_f \left( \mathbb{E}_{Y_0 \sim q_T} \left[ \| \hat{Y}_{t_{k+1}}^{\phi} - Y_{t'_{k+1}} \|_2^2 \right] \right)^{1/2}
$$
$$
\leq \sum_{k=0}^{K-1} L_f g(T-t)^2 h_k'^2 \left( \frac{R^3}{(T-t'_k)^5} + \frac{R^2 \sqrt{d}}{(T-t'_k)^4} \right)
$$
$$
\leq \frac{L_f R^2 (R + \sqrt{d})}{\delta^2} \sum_{k=0}^{K-1} \frac{h_k'^2}{(T-t'_k)^2} \ .
$$

We know that

$$
\sum_{k=1}^{K} \frac{h_k^2}{t_k^2} \asymp h \sum_{k=1}^{K} \frac{h_k}{t_k^{\frac{a+1}{a}}} \asymp h \int_{\delta}^{T} \frac{1}{t^{\frac{a+1}{a}}} \, \mathrm{d}t \asymp h \delta^{-\frac{1}{a}} \asymp \frac{(T/\delta)^{\frac{1}{a}}}{K} \ ,
$$

which has a better dependence on $T$ and $\delta$ compared with Eq. 10 and achieve an improved result $L_f / \epsilon_{W_2}^{3+2/a}$.

*Remark* 4.15. To make a clearer discussion on the role of VESDE, we choose a uniform stepsize ($a = 1$), which is used by Lyu et al. (2024). Under this setting, the difference between our work and Lyu et al. (2024) is the forward process. Lyu et al. (2024) use the VPSDE forward process and achieve $L_f / (\epsilon_{W_2} \delta^3)$ result. We adopt VESDE forward process and achieve $L_f T / (\epsilon_{W_2} \delta^3) = L_f / (\epsilon_{W_2}^2 \delta^3)$ [6]. Under the uniform stepsize, our better result is due to better $\delta$ of VESDE. More specifically, to guarantee $W_2(q_\delta, q_0) \leq \epsilon_{W_2}$ (the third term of Eq. 7), we require $\delta$ has order of $\epsilon_{W_2}$ for VESDE and $\epsilon_{W_2}^2$ for VPSDE. Hence, Lyu et al. (2024) achieve $L_f / \epsilon_{W_2}^7$ result for VPSDE and we achieve $L_f / \epsilon_{W_2}^5$ for VESDE (uniform stepsize setting).

## 5. Conclusion

In this work, we make the first step to explain why consistency models perform well from a theoretical perspective. More specifically, we bridge the gap between theory and application of consistency models by analyzing the discretization complexity of the consistency model with VESDE forward process and EDM discretization scheme. Under this realistic setting with great empirical performance, we first show the state-of-the-art discretization complexity $O\left( L_f / \epsilon_{W_2}^{3+\frac{2}{a}} \right)$ by making full use of time-dependent score perturbation lemma. Then, we analyze the 2-step sampling

---

algorithm proposed by Song et al. (2023) and show that this algorithm further improves discretization complexity to $O\left( L_f / \epsilon_{W_2}^{3+\frac{3}{2a}} \right)$, which explain the widely used of multi-step sampling algorithm in application.

In conclusion, we achieve competitive results for consistency models compared with diffusion models and show the potential of consistency models.

**Future work.** In this work, we directly assume the score and consistency function are accurate enough. For the score function, some works analyze it learning process Chen et al. (2023b); Yuan et al. (2023). For the consistency function, Dou et al. (2024) analyze its estimation error. However, as discussed in Sec. 3.2, they run multi-step PFODE instead of one-step PFODE, which is time-consuming and does not match the empirical operation. Hence, it is interesting to analyze the learning process of consistency models under a realistic setting and achieve an end-to-end analysis. Very recently, Lu and Song (2024) train consistency models in a continuous time and achieve a great performance. Since the continuous time models use $\frac{\mathrm{d} f_{\theta^-}(X_t, t)}{\mathrm{d}t}$ instead of $f_{\theta^-}(X_{t-\Delta t}, t - \Delta t)$ ($\Delta t$ is $h_{k+1} - h_k$ in our work. Here we use the uniform stepsize for convenience), there are not well-defined discretization complexity $K = T/\Delta t$ for continuous-time models, however, due to the absence of $\Delta t$, the training process of continuous time models is less stable than discrete time consistency models, which is the core problem for continuous time models. Hence, it is also an interesting future work to analyze how to make the training process of continuous time consistency models more stable from the theoretical perspective.

## Impact Statement

Our work aims to deepen the understanding of consistency models from the theoretical perspective. Since consistency models are a new class of one-step generative models, the societal impact is similar to general generative models (Mirsky and Lee, 2021).

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

# Appendix

## A. Notation

We denote by $W_1$ and $W_2$ the Wasserstein distance of order one and two, respectively. Note that $W_1$ guarantee is weaker than $W_2$ guarantee since $W_1(p, q) \leq W_2(p, q)$. The push-forward operator $\sharp$ is associated with a measurable map $f : \mathcal{M}' \to \mathcal{N}$. For any measure $\mu$ over $\mathcal{M}'$, we define the push-forward measure $f\sharp\mu$ over $\mathcal{N}$ by: $f\sharp\mu(A) = \mu\left(f^{-1}(A)\right)$, for any $A$ be measurable set in $\mathcal{N}$.

Before introducing our theoretical guarantee, we first introduce the notation for the diffusion and consistency models. Let $q_0$ be the data distribution.

Diffusion models:

- Let $(X_t)_{t \in [0,T]}$ be the random variable of the forward process (Equation (1)). We define by $\delta = t_0 \leq t_1 \leq \cdots \leq t_K = T$ and $h_k = t_k - t_{k-1}$ the discretization points and stepsize.

- Let $t' = T - t$ be the reverse time and $(Y_{t'})_{t' \in [0,T]} = (X_{T-t'})_{t' \in [0,T]}$ be the random variable of reverse process (Equation (2)). We define by $t'_k = T - t_{K-k}$ and $h'_k = h_{K-k}$ the discretization points and stepsize in the reverse process.

- Let $p_K$ be the distribution generated by running the discrete PFODE process (the last equation of Section 3.1) with $s_\phi$, the complexity of the sample is the requirement of $K$ to guarantee $W_2(p_K, q_0) \leq \epsilon_{W_2}$.

Consistency models:

- The goal of consistency models is to learn a consistency function $\boldsymbol{f_\theta}(Y, t')$ to directly map pure noise $Y \sim \mathcal{N}(0, T^2 I_d)$ and $t' = 0$ (the start of the reverse process) to $q_0$.

- We denote by $\boldsymbol{f_{\theta,0}}\sharp\mathcal{N}(\mathbf{0}, T^2 I_d)$ the generated distribution of the above operation. Since the consistency function is one step, the discretization complexity is the requirement of $K$ in the training process (Equation (4)) to guarantee $W_2(\boldsymbol{f_{\theta,0}}\sharp\mathcal{N}(\mathbf{0}, T^2 I_d), q_0) \leq \epsilon_{W_2}$.

## B. The Analysis for Consistency Model

**Theorem 4.7.** *Assume Assumption 4.1, 4.2, 4.3, 4.4 holds and consider the EDM stepsize (5). Then, one-step generation error $W_2\left(\boldsymbol{f_{\theta,0}}\sharp\mathcal{N}(0, T^2 I_d), q_0\right)$ is bounded by*

$$\frac{R^2}{T} + \frac{L_f R^2 (R + \sqrt{d})(T/\delta)^{\frac{1}{a}}}{K\delta^2} + L_f T \epsilon_{\text{score}} + T \epsilon_{\text{cm}} + \sqrt{d}\delta .$$

*Furthermore, by choosing $T \geq R^2/\epsilon_{W_2}$, $\delta = \epsilon_{W_2}/\sqrt{d}$, $\epsilon_{\text{cm}} \leq \epsilon_{W_2}/T$ and $\epsilon_{\text{score}} \leq \epsilon_{W_2}^2/(L_f R^2)$, the output is $\epsilon_{W_2}$-close to $q_0$ with discretization complexity*

$$K = O\left(\frac{L_f R^{2 + \frac{2}{a}}(R + \sqrt{d})d^{1 + \frac{1}{2a}}}{\epsilon_{W_2}^{3 + \frac{2}{a}}}\right) .$$

**Proof.** We first decompose $W_2\left(\boldsymbol{f_{\theta,0}}\sharp\mathcal{N}(\mathbf{0}, T^2 I_d), q_0\right)$:

$$W_2\left(\boldsymbol{f_{\theta,0}}\sharp\mathcal{N}(\mathbf{0}, T^2 I_d), q_0\right) \leq W_2\left(\boldsymbol{f_{\theta,0}}\sharp\mathcal{N}(\mathbf{0}, T^2 I_d), \boldsymbol{f_{\theta,0}}\sharp q_T\right) + W_2\left(\boldsymbol{f_{\theta,0}}\sharp q_T, q_\delta\right) + W_2\left(q_\delta, q_0\right) ,$$

where the first term is the reverse beginning error due to the forward process, the second term is the discretization error due to the discretization training, and the third term is due to the early stopping technique. We first define a joint distribution $\gamma \in \Gamma\left(\mathcal{N}(\mathbf{0}, T^2 I_d), q_T\right)$ between $\mathcal{N}(\mathbf{0}, T^2 I_d)$ and $q_T$ and take a couple of $(\bar{Y}_0, Y_0) \sim \gamma$, which indicates

$$\int_{\mathbb{R}^d} \gamma(\cdot, Y_0)\mathrm{d}Y_0 = \mathcal{N}(\mathbf{0}, T^2 I_d)$$

$$\int_{\mathbb{R}^d} \gamma(\bar{Y}_0, \cdot)\mathrm{d}\bar{Y}_0 = q_T .$$

Then, we have that

$$W_2\left(\boldsymbol{f_{\theta,0}}\sharp\mathcal{N}\left(\boldsymbol{0},T^2 I_d\right),q_\delta\right)$$

$$\leq \left(\mathbb{E}_\gamma\left[\left\|\boldsymbol{f_\theta}\left(\bar{Y}_0,0\right)-\boldsymbol{f^v}\left(Y_0,0\right)\right\|_2^2\right]\right)^{1/2}$$

$$\leq \left(\mathbb{E}_\gamma\left[\left\|\boldsymbol{f_\theta}\left(\bar{Y}_0,0\right)-\boldsymbol{f_\theta}\left(Y_0,0\right)\right\|_2^2\right]\right)^{1/2}+\left(\mathbb{E}_\gamma\left[\left\|\boldsymbol{f_\theta}\left(Y_0,0\right)-\boldsymbol{f^v}\left(Y_0,0\right)\right\|_2^2\right]\right)^{1/2}$$

$$\leq L_{f,0}\left(\mathbb{E}_\gamma\left[\left\|\bar{Y}_0-Y_0\right\|_2^2\right]\right)^{1/2}+\left(\mathbb{E}_\gamma\left[\left\|\boldsymbol{f_\theta}\left(Y_0,0\right)-\boldsymbol{f^v}\left(Y_0,0\right)\right\|_2^2\right]\right)^{1/2}$$

$$\leq \frac{R}{T}W_2(\mathcal{N}(0,T^2 I_d),q_T)+\left(\mathbb{E}_\gamma\left[\left\|\boldsymbol{f_\theta}\left(Y_0,0\right)-\boldsymbol{f^v}\left(Y_0,0\right)\right\|_2^2\right]\right)^{1/2}$$

$$= \frac{R}{T}W_2(\mathcal{N}(0,T^2 I_d),q_T)+\left(\mathbb{E}_{Y_0\sim q_T}\left[\left\|\boldsymbol{f_\theta}\left(Y_0,0\right)-\boldsymbol{f^v}\left(Y_0,0\right)\right\|_2^2\right]\right)\,.$$

where the first inequality follow the fact $\boldsymbol{f^v}\left(Y_0,0\right)=X_\delta$ when $Y_0\sim q_T$ and the last inequality follows the fact that $\gamma$ can be any coupling between $\mathcal{N}(0,T^2 I_d)$ and $q_T$.

**The reverse beginning error.** We first control the reverse beginning error term. When considering the $W_2$ guarantee, we have that

$$W_2(\mathcal{N}(0,T^2 I_d),q_T)\leq \left(\mathbb{E}_{q_0}\left[\left\|X_0+(T-T)\,\xi\right\|_2^2\right]\right)^{1/2}\leq R\,.$$

**The discretization error.** In this section, we control the discretization error:

$$\left(\mathbb{E}_{Y_0\sim q_T}\left[\left\|\boldsymbol{f_\theta}\left(Y_0,0\right)-\boldsymbol{f^v}\left(Y_0,0\right)\right\|_2^2\right]\right)^{1/2}$$

$$= \left(\mathbb{E}_{Y_0\sim q_T}\left[\left\|\sum_{k=0}^{K-1}\left(\boldsymbol{f_\theta}\left(Y_{t'_k},t'_k\right)-\boldsymbol{f_\theta}\left(Y_{t'_{k+1}},t'_{k+1}\right)\right)\right\|_2^2\right]\right)^{1/2}$$

$$\leq \sum_{k=0}^{K-1}\left(\mathbb{E}_{Y_0\sim q_T}\left[\left\|\boldsymbol{f_\theta}\left(Y_{t'_k},t'_k\right)-\boldsymbol{f_\theta}\left(Y_{t'_{k+1}},t'_{k+1}\right)\right\|_2^2\right]\right)^{1/2}$$

$$= \sum_{k=0}^{K-1}\left(\mathbb{E}_{Y_0\sim q_T}\left[\left\|\boldsymbol{f_\theta}\left(Y_{t'_k},t'_k\right)-\boldsymbol{f_\theta}\left(\hat{Y}^\phi_{t'_{k+1}},t'_{k+1}\right)+\boldsymbol{f_\theta}\left(\hat{Y}^\phi_{t'_{k+1}},t'_{k+1}\right)-\boldsymbol{f_\theta}\left(Y_{t'_{k+1}},t'_{k+1}\right)\right\|_2^2\right]\right)^{1/2}$$

$$\leq \sum_{k=0}^{K-1}\left(\mathbb{E}_{Y_0\sim q_T}\left[\left\|\boldsymbol{f_\theta}\left(Y_{t'_k},t'_k\right)-\boldsymbol{f_\theta}\left(\hat{Y}^\phi_{t'_{k+1}},t'_{k+1}\right)\right\|_2^2\right]\right)^{1/2}$$

$$+\sum_{k=0}^{K-1}\left(\mathbb{E}_{Y_0\sim q_T}\left[\left\|\boldsymbol{f_\theta}\left(\hat{Y}^\phi_{t'_{k+1}},t'_{k+1}\right)-\boldsymbol{f_\theta}\left(Y_{t'_{k+1}},t'_{k+1}\right)\right\|_2^2\right]\right)^{1/2}:=E_1+E_2\,,$$

where the first inequality follows the fact that $\boldsymbol{f^{ex}}\left(Y_0,0\right)=X_\delta=\boldsymbol{f_\theta}\left(Y_{T-\delta},T-\delta\right)$. For term $E_1$, since we assume the learned consistency model is accurate enough (Assumption 4.3), then we have that:

$$E_1=\sum_{k=0}^{K-1}\left(\mathbb{E}_{Y_0\sim q_T}\left[\left\|\boldsymbol{f_\theta}\left(Y_{t'_k},t'_k\right)-\boldsymbol{f_\theta}\left(\hat{Y}^\phi_{t'_{k+1}},t'_{k+1}\right)\right\|_2^2\right]\right)^{1/2}$$

$$= \sum_{k=0}^{K-1}\left(\mathbb{E}_{Y_{t'_k}\sim q_{t'_k}}\left[\left\|\boldsymbol{f_\theta}\left(Y_{t'_k},t'_k\right)-\boldsymbol{f_\theta}\left(\hat{Y}^\phi_{t'_{k+1}},t'_{k+1}\right)\right\|_2^2\right]\right)^{1/2}$$

$$\leq \epsilon_{cm}\sum_{k=0}^{K-1}h'_k=\epsilon_{cm}\left(T-\delta\right)\,.$$

For term $E_2$, we know that

$$
\begin{aligned}
E_2 &= \sum_{k=0}^{K-1} \left( \mathbb{E}_{Y_0 \sim q_T} \left[ \left\| \boldsymbol{f_\theta} \left( \hat{Y}_{t'_{k+1}}^\phi, t'_{k+1} \right) - \boldsymbol{f_\theta} \left( Y_{t'_{k+1}}, t'_{k+1} \right) \right\|_2^2 \right] \right)^{1/2} \\
&\leq \sum_{k=0}^{K-1} L_f \left( \mathbb{E}_{Y_0 \sim q_T} \left[ \left\| \hat{Y}_{t'_{k+1}}^\phi - Y_{t'_{k+1}} \right\|_2^2 \right] \right)^{1/2} \\
&= \sum_{k=0}^{K-1} L_f \left( \mathbb{E}_{Y_{t'_k} \sim q_{t'_k}} \left[ \left\| \hat{Y}_{t'_{k+1}}^\phi - Y_{t'_{k+1}} \right\|_2^2 \right] \right)^{1/2} \\
&\lesssim \sum_{k=0}^{K-1} L_f \left( h_k'^2 \left( \frac{R^3}{(T-t'_k)^4} + \frac{R^2\sqrt{d}}{(T-t'_k)^3} \right) + h_k' \epsilon_{\text{score}} \right),
\end{aligned}
$$

where the last inequality comes from Lemma C.2. When considering EDM stepsize

$$
t_k = (\delta + kh)^a, \quad h = \frac{T^{\frac{1}{a}} - \delta}{K},
$$

we know that $\frac{h_k}{h} \asymp t_k^{\frac{a-1}{a}}$,

$$
\sum_{k=1}^{K} \frac{h_k^2}{t_k^2} \asymp h \sum_{k=1}^{K} \frac{h_k}{t_k^{\frac{a+1}{a}}} \asymp h \int_\delta^T \frac{1}{t^{\frac{a+1}{a}}} \, dt \asymp h\delta^{-\frac{1}{a}} \asymp \frac{(T/\delta)^{\frac{1}{a}}}{K} .
$$

Then, term $E_2$ is control by the following inequality:

$$
\begin{aligned}
E_2 &\leq \sum_{k=1}^{K} L_f \left( \frac{R^2(R+\sqrt{d})h_k^2}{t_k^2 \delta^2} + h_k \epsilon_{\text{score}} \right) \\
&\leq \frac{L_f R^2 (R+\sqrt{d})(T/\delta)^{\frac{1}{a}}}{K\delta^2} + L_f T \epsilon_{\text{score}}
\end{aligned}
$$

For term $W_2(q_\delta, q_0)$, by using Lemma E.3, it is smaller than $\sqrt{d}\sigma_\delta = \sqrt{d}\delta$.

Combined with the reverse beginning, discretization and early stopping term, we have that

$$
W_2 \left( \boldsymbol{f_{\theta,0}} \sharp \mathcal{N} \left( \boldsymbol{0}, T^2 I_d \right), q_0 \right) \lesssim \frac{R^2}{T} + \frac{L_f R^2 (R+\sqrt{d})(T/\delta)^{\frac{1}{a}}}{K\delta^2} + L_f T \epsilon_{\text{score}} + \epsilon_{\text{cm}} T + \sqrt{d}\delta .
$$

To make the above inequality smaller than $\epsilon_{W_2}$, we choose $T \geq R^2/\epsilon_{W_2}$, $\delta = \epsilon_{W_2}/\sqrt{d}$, $\epsilon_{\text{cm}} \leq \epsilon_{W_2}/T$, and guarantee

$$
\frac{L_f R^2 (R+\sqrt{d}) T^{\frac{1}{a}}}{K\delta^{2+\frac{1}{a}}} \leq \epsilon_{W_2} ,
$$

which indicates that $K \geq \frac{L_f R^{2+\frac{2}{a}}(R+\sqrt{d})d^{1+\frac{1}{2a}}}{\epsilon_{W_2}^{3+\frac{2}{a}}}$. After determining the discretization complexity $K$, we can also obtain the requirement of the approximated score function $\epsilon_{\text{score}} \leq \epsilon_{W_2}^2/(L_f R^2)$. ∎

**Corollary B.1.** *Assume Assumption 4.1, 4.2, 4.3, 4.4 holds and consider the exponential decay stepsize $h_k = rt_k$, where $r = \epsilon_{W_2}^3 / \left( R^2 \log^2(T/\delta) \right)$. Following $T, \delta, \epsilon_{\text{cm}}$ in Thm. 4.7 and choosing $\epsilon_{\text{score}} \leq \epsilon_{W_2}\delta/(R^2 \log(T/\delta))$, the output is $\epsilon_{W_2}$-close to $q_0$ with discretization complexity*

$$
K = O \left( L_f R^2 d(R+\sqrt{d}) \log^2(T/\delta)/\epsilon_{W_2}^3 \right) .
$$

**Proof**. For the theoretical-friendly exponential decay stepsize, we have that

$$E_2 \leq L_f \sum_{k=1}^{K=\frac{1}{r}\log(T/\delta)} \left( \frac{R^2(R+\sqrt{d})h_k^2}{t_k^2\delta^2} + h_k\epsilon_{\text{score}} \right)$$

$$\lesssim \frac{L_f R^2(R+\sqrt{d})\log^2(T/\delta)}{K\delta^2} + L_f T\epsilon_{\text{score}}.$$

Combined with the reverse beginning term and discretization term, we know that

$$W_2\left(\boldsymbol{f}_{\boldsymbol{\theta},0}\sharp\mathcal{N}\left(\boldsymbol{0},T^2 I_d\right),q_0\right) \lesssim \frac{R^2}{T} + \frac{L_f R^2(R+\sqrt{d})\log^2(T/\delta)}{K\delta^2} + L_f T\epsilon_{\text{score}} + \epsilon_{\text{cm}}T + \sqrt{d}\delta.$$

In order to guarantee the right hand of above inequality smaller than $\epsilon_{W_2}$, we choose $T \geq R^2/\epsilon_{W_2}$, $\delta = \epsilon_{W_2}/\sqrt{d}$, $\epsilon_{\text{cm}}^2 \leq \epsilon_{W_2}^2/T^2$, and guarantee

$$\frac{R^2(R+\sqrt{d})\log^2(T/\delta)}{K\delta^2} \leq \epsilon_{W_2},$$

which indicates the discretizaiton complexity is $K \geq \frac{R^2 d(R+\sqrt{d})\log^2(T/\delta)}{\epsilon_{W_2}^3}$. After obtaining the requirement of $K$, we can also obtain the requirement of approximated score function $\epsilon_{\text{score}} \leq \epsilon_{W_2}^2/(L_f R^2)$. ∎

At the end of this part, we provide the proof of multi-step sampling.

**Corollary B.2.** *Assume Assumption 4.1, 4.2, 4.3, 4.4 holds and $L_{f,T/2} = \Theta(R/T)$. Choosing EDM stepsize and $\tau_2 = T/2$ (2-step sampling), $W_2(p_2, q_0)$ is bounded by*

$$\sqrt{d}\delta + R^3/(\tau_2 T) + L_f R^2(R+\sqrt{d})(T/\delta)^{\frac{1}{a}}/(K\delta^2)$$
$$+ (RT/\tau_2 + \tau_2)(L_f\epsilon_{\text{score}} + \epsilon_{\text{cm}}).$$

*By choosing $T \geq R^{1.5}/\sqrt{\epsilon_{W_2}}$, $\delta = \epsilon_{W_2}/\sqrt{d}$, $\epsilon_{\text{cm}} \leq \epsilon_{W_2}/T$ and $\epsilon_{\text{score}} \leq \epsilon_{W_2}^2/(L_f R^2)$, the output is $\epsilon_{W_2}$-close to $q_0$ with discretization complexity*

$$K = O\left(L_f R^{2+\frac{3}{2a}}(R+\sqrt{d})d^{1+\frac{1}{2a}}/\epsilon_{W_2}^{3+\frac{3}{2a}}\right).$$

**Proof**. Take a couple of $(\boldsymbol{Y}, \boldsymbol{Z}) \sim \gamma(\boldsymbol{y}, \boldsymbol{z})$ where $\gamma \in \Gamma(p_{n-1}, q_\delta)$, take $\boldsymbol{\xi} \sim \mathcal{N}(\boldsymbol{0}, \boldsymbol{I}_d)$, then we have

$$\hat{\boldsymbol{Y}} = \boldsymbol{Y} + \tau_n\boldsymbol{\xi} \sim \mu_n,$$
$$\hat{\boldsymbol{Z}} = \boldsymbol{Z} + \tau_n\boldsymbol{\xi} \sim q_{\tau_n}.$$

Similar with Corollary 10 of Lyu et al. (2024), we have that

$$W_2(p_2, q_0)$$

$$\lesssim W_2(q_\delta, q_0) + L_{f,T-\tau_2}W_2(\mu_n, q_{\tau_2}) + \frac{L_f R^2(R+\sqrt{d})(\tau_2/\delta)^{\frac{1}{a}}}{K\delta^2} + L_f\tau_2\epsilon_{\text{score}} + \epsilon_{\text{cm}}\tau_2$$

$$\lesssim W_2(q_\delta, q_0) + L_{f,T-\tau_2}\left(\mathbb{E}_\gamma\|\hat{\boldsymbol{Y}} - \hat{\boldsymbol{Z}}\|_2^2\right)^{1/2} + \frac{L_f R^2(R+\sqrt{d})(\tau_2/\delta)^{\frac{1}{a}}}{K\delta^2} + L_f\tau_2\epsilon_{\text{score}} + \epsilon_{\text{cm}}\tau_2$$

$$\lesssim W_2(q_\delta, q_0) + L_{f,T-\tau_2}\left(\mathbb{E}_\gamma\|\boldsymbol{Y} - \boldsymbol{Z}\|_2^2\right)^{1/2} + \frac{L_f R^2(R+\sqrt{d})(\tau_2/\delta)^{\frac{1}{a}}}{K\delta^2} + L_f\tau_2\epsilon_{\text{score}} + \epsilon_{\text{cm}}\tau_2$$

$$\lesssim \sqrt{d}\delta + \frac{R}{\tau_2}\left(\frac{R^2}{T} + \frac{L_f R^2(R+\sqrt{d})(T/\delta)^{\frac{1}{a}}}{K\delta^2} + L_f T\epsilon_{\text{score}} + \epsilon_{\text{cm}}T\right)$$

$$+ \frac{L_f R^2(R+\sqrt{d})(\tau_2/\delta)^{\frac{1}{a}}}{K\delta^2} + L_f\tau_2\epsilon_{\text{score}} + \epsilon_{\text{cm}}\tau_2$$

where the first line of the last inequality is introduced by the previous sampling process, and the remaining term is the discretization error of this phase.

∎

## C. The Error of One Step PFODE for the VESDE Forward Process

As a beginning, we control $\mathbb{E}_{X_t \sim q_t} [\|\partial_t \nabla \log q_t (X_t)\|_2]$. In previous work, the following lemma is named the score perturbation lemma, which depends on the uniform Lipschitz constant $L_{\text{score}}$ for the score function. As discussed in Section 4.4, a time-dependent results is important for better results for the EDM and exponential decay stepsize.

**Lemma C.1.** *For VESDE (Eq. 1), we have that*

$$\mathbb{E}_{X_t \sim q_t} [\|\partial_t \nabla \log q_t (X_t)\|_2] \leq R^3/t^5 + R^2\sqrt{d}/t^4 .$$

**Proof.** For the VESDE forward process, we know that (here, we use the forward process notation):

$$\dot{X}_t = -t\nabla \log q_t (X_t) ,$$

which indicates

$$\partial_t \nabla \log q_t (X_t) = [\partial_t \nabla \log q_t(y)]|_{y=X_t} - t\nabla^2 \log q_t (X_t) \nabla \log q_t (X_t)$$

Hence, the following inequality holds

$$
\begin{aligned}
\|\partial_t \nabla \log q_t (X_t)\| &\leq \| [\partial_t \nabla \log q_t(y)]|_{y=X_t} \|_2 + t\|\nabla^2 \log q_t (X_t)\|_2 \|\nabla \log q_t(X_2)\|_2 \\
&\leq \frac{R^3 + R^2\|X_t\|_2}{t^5} + t\frac{R^2}{t^4}\frac{\|X_t\|_2 + R}{t^2} \\
&\leq \frac{R^3}{t^5} + \frac{R^2\sqrt{d}}{t^4} ,
\end{aligned}
$$

where the second inequality follows Lemma E.1 and Lemma E.2 and the last inequality follows Lemma E.4. ∎

When considering the consistency distillation training paradigm, we need to run a one-step reverse PFODE starting from $Y_{t'_k}$ to obtain $\widehat{Y}^\phi_{t'_{k+1}}$. Hence, we need to control one step starting from the same distribution $q$.

**Lemma C.2.** *Suppose Assumption 4.1 and Assumption 4.2 hold and assuming $\frac{\sigma_s^2 - \sigma_t^2}{\sigma_t^2} \leq \frac{1}{2d}$ for any $0 \leq t \leq s \leq T$, then for the small interval $t \in [t'_k, t'_{k+1}]$ for $\forall k \in [0, K-1]$, we have that*

$$W_2^2 \left( qQ^{t'_k, h'_k}_{\text{ODE}}, q\widehat{P}^{t'_k, h'_k}_{\text{ODE}} \right) \lesssim h'^4_k \left( \frac{R^6}{(T-t'_k)^8} + \frac{R^4 d}{(T-t'_k)^6} \right) + h'^2_k \epsilon^2_{\text{score}} .$$

*where $q\widehat{P}^{t'_k}_{\text{ODE}}$ means the output of one-step PFODE with fixed approximated score starting from $q$.*

**Proof.** For $t \in [t'_k, t'_{k+1}]$, the reverse PFODE is

$$
\begin{aligned}
\dot{Y}_t &= \frac{g(T-t)^2}{2}\nabla \log q_{T-t} (Y_t) , \\
\dot{\widehat{Y}}_t &= \frac{g(T-t)^2}{2}s_\phi \left( \widehat{Y}_{t'_k}, T - t'_k \right) ,
\end{aligned}
$$

for $t'_k \leq t \leq t'_{k+1}$, with $Y_{t'_k} = \widehat{Y}_{t'_k} \sim q, Y_{t'_k+h'_k} \sim qQ_{\text{ODE}}$, and $\widehat{Y}_{t'_k+h'_k} \sim q\widehat{P}_{\text{ODE}}$. Then, we have that

$$
\begin{aligned}
\partial_t \left\| Y_t - \widehat{Y}_t \right\|^2 &= 2\left\langle Y_t - \widehat{Y}_t, \dot{Y}_t - \dot{\widehat{Y}}_t \right\rangle \\
&= 2\left\langle Y_t - \widehat{Y}_t, \frac{g(T-t)^2}{2}\left( \nabla \log q_{T-t} (Y_t) - s_\phi \left( \widehat{Y}_{t'_k}, T - t'_k \right) \right) \right\rangle \\
&\leq \frac{1}{h'_k}\left\| Y_t - \widehat{Y}_t \right\|^2_2 + \frac{h'_k g(T-t)^4}{4}\left\| \nabla \log q_{T-t} (Y_t) - s_\phi \left( \widehat{Y}_{t'_k}, T - t'_k \right) \right\|^2_2
\end{aligned}
$$

As the next step, we use the Grönwall's inequality to control the one-step discretization error.

**Grönwall's inequality.** Let $\eta(\cdot)$ be a nonnegative, absolutely continuous function on $[0, h_k']$, which satisfies for a.e. $t$ the differential inequality

$$\eta'(t) \le \phi(t)\eta(t) + \psi(t)$$

where $\phi(t)$ and $\psi(t)$ are nonnegative, summable function on $[0, h_k']$. Then

$$\eta(t) \le e^{\int_0^t \phi(s)\mathrm{d}s} \left[ \eta(0) + \int_0^t \psi(s)\mathrm{d}s \right] .$$

By setting $\eta(t) = \left\| Y_t - \widehat{Y}_t \right\|_2^2$ and $\psi(t) = \frac{h_k' g(T-t)^4}{4} \left\| \nabla \log q_{T-t}(Y_t) - s_\phi\left(\widehat{Y}_{t_k'}, T - t_k'\right) \right\|_2^2$. We note that $Y_{t_k'} = \widehat{Y}_{t_k'} \sim q$ starts from the same distribution, $\eta(0) = 0$. Then, we can obtain that

$$
\begin{aligned}
&\mathbb{E}\left[ \left\| Y_{t_k' + h_k'} - \widehat{Y}_{t_k' + h_k'} \right\|_2^2 \right] \\
&\le \exp\left( \int_0^{h_k'} \frac{1}{h_k'} \mathrm{d}t \right) \int_{t_k'}^{t_k' + h_k'} \frac{g(T-t)^4 h_k'}{4} \mathbb{E}\left[ \left\| \nabla \log q_{T-t}(Y_t) - s_\phi\left(\widehat{Y}_{t_k'}, T - t_k'\right) \right\|_2^2 \right] \mathrm{d}t \\
&\le \int_{t_k'}^{t_k' + h_k'} g(T-t)^4 h_k' \mathbb{E}\left[ \left\| \nabla \log q_{T-t}(Y_t) - s_\phi\left(\widehat{Y}_{t_k'}, T - t_k'\right) \right\|_2^2 \right] \mathrm{d}t ,
\end{aligned}
$$

Then, by using Lemma 4.14, we have that

$$
\begin{aligned}
&\mathbb{E}\left[ \left\| Y_{t_k' + h_k'} - \widehat{Y}_{t_k' + h_k'} \right\|_2^2 \right] \\
&\le \int_{t_k'}^{t_k' + h_k'} g(T-t)^4 h_k' \mathbb{E}\left[ \left\| \nabla \log q_{T-t}(Y_t) - s_\phi\left(\widehat{Y}_{t_k'}, T - t_k'\right) \right\|_2^2 \right] \mathrm{d}t \\
&\lesssim \int_{t_k'}^{t_k' + h_k'} g(T-t)^4 h_k' \left[ \left\| \nabla \log q_{T-t}(Y_t) - \nabla \log_{T-t_k'}\left(\widehat{Y}_{t_k'}\right) \right\|_2^2 \right] + \frac{h_k' g(T-t)^4 \epsilon_{\mathrm{score}}^2}{\sigma_{T-t}^2} \mathrm{d}t \\
&\lesssim \int_{t_k'}^{t_k' + h_k'} g(T-t)^4 h_k'^3 \left[ \left\| \partial_t \nabla \log q_{T-t}(Y_t) \right\|_2^2 \right] \mathrm{d}t + h_k'^2 \epsilon_{\mathrm{score}}^2 \\
&\lesssim h_k'^4 \left( \frac{R^6}{(T - t_k')^8} + \frac{R^4 d}{(T - t_k')^6} \right) + h_k'^2 \epsilon_{\mathrm{score}}^2 ,
\end{aligned}
$$

where the last inequality follows Lemma 4.14.

$\blacksquare$

# D. The Discussion on the Previous Work

## D.1. The Linear Solution Trajectory of VESDE ($\sigma_t^2 = t^2$)

In this part, we show why it is possible for VESDE ($\sigma_t^2 = t^2$) to generate samples with a single Euler step. In this part, we use the forward process notation $X_t$ and $t$ to keep consistent with empirical works (Karras et al., 2022; Song et al., 2023). We first provide the exact form of $\nabla \log q_t(\cdot)$. As shown in Karras et al. (2022) and Benton et al. (2024), we have that

$$\nabla \log q_t(X_t) = \frac{\mathbb{E}[X_0 \mid X_t] - X_t}{t^2} ,$$

where $\mathbb{E}[X_0 \mid X_t]$ is the posterior mean given $X_t$.

By choosing VESDE ($\sigma_t^2 = t^2$) as the forward process, the reverse process becomes [7]

$$\mathrm{d}X_t = -t\nabla \log q_t\left(X_t\right)\mathrm{d}t, X_T \sim q_T\,.$$

With the exact form of $\nabla \log q_t(\cdot)$, we have that

$$\mathrm{d}X_t = \frac{X_t - \mathbb{E}\left[X_0 \mid X_t\right]}{t}\mathrm{d}t, X_T \sim q_T\,.$$

Then, starting from $X_T$, if doing one-step Euler, we have that (we denote the generated samples as $\bar{X}_0$)

$$\bar{X}_0 = X_T + (0 - T)\frac{X_T - \mathbb{E}\left[X_0 \mid X_T\right]}{T} = \mathbb{E}\left[X_0 \mid X_T\right]\,.$$

As discussed in Karras et al. (2022) and Liu et al. (2023a), a linear trajectory indicates it is possible to generate samples with a single Euler step and is the basis of a one-step model. Hence, consistency models adopt VESDE ($\sigma_t^2 = t^2$) as the forward process (Song et al., 2023; Song and Dhariwal, 2023).

### D.2. The Detail Calculation of Previous Consistency Models Results

In this part, we show current discretization complexity results and discuss the reason why the noise schedule of Li et al. (2024a) relies heavily on the VPSDE forward process.

**The results of Lyu et al. (2024).** As shown in Corollary 8 of Lyu et al. (2024), the discretization complexity of consistency distillation (under the bounded support) is

$$\widetilde{O}\left(\frac{d^{1/2}R^3\left(R^6 \vee d^3\right)L_f}{\epsilon_{W_2}^7}\right)\,.$$

**The results of Li et al. (2024a).** In this part, we first show the noise schedule of Li et al. (2024a) and discuss the reason why this schedule depends heavily on VPSDE. Li et al. (2024a) describe the VPSDE in a discrete perspective instead of a continuous forward process:

$$X_0 \sim q_0,$$
$$X_k = \sqrt{1 - \beta_k}X_{k-1} + \sqrt{\beta_t}B_k, \quad 1 \leqslant k \leqslant K\,.$$

Let

$$\alpha_k := 1 - \beta_k, \quad \bar{\alpha}_k := \prod_{k'=1}^{k} \alpha_{k'}, \quad 1 \leqslant k \leqslant K\,.$$

Then, we know that $X_k = \sqrt{\bar{\alpha}_k}X_0 + \sqrt{1 - \bar{\alpha}_k}\bar{W}_k$ for some $\bar{W}_k \sim \mathcal{N}\left(0, I_d\right)$, which indicates $X_K$ is approximately $\mathcal{N}\left(0, I_d\right)$ (VPSDE) with suitable noise schedule. Li et al. (2024a) choose a specific noise schedule

$$\beta_1 = 1 - \alpha_1 = \frac{1}{K^{c_0}}\,,$$
$$\beta_k = 1 - \alpha_k = \frac{c_1 \log K}{K}\min\left\{\beta_1\left(1 + \frac{c_1 \log K}{K}\right)^k, 1\right\}, \quad 2 \leqslant k \leqslant K\,,$$

where $c_0, c_1 > 0$ are large enough numerical constants. We note that when $K$ goes to $+\infty$, $\beta_k$ will goes to 0. It is quietly different from VESDE forward process since $\sigma_t^2 = t^2$ would goes to $+\infty$ when $T$ goes to $+\infty$ ($K \to +\infty$). Hence, this noise schedule heavily depends on the form of VPSDE.

With the above specific noise schedule for VPSDE, Li et al. (2024a) show the discretization complexity of consistency training paradigm with VPSDE forward process:

$$K = \widetilde{O}\left(\frac{L_f^3 d^{5/2}}{\epsilon_{W_1}}\right)\,.$$

---

[7]Note that the coefficient of the above PFODE is negative, which is slightly different with Eq. 2. This is because we use the forward process notation and the generation timeline is $X_T \to X_0$.

**The results of Dou et al. (2024).**

(a) Consistency Distillation Paradigm.

As shown in Theorem 4.1 of Dou et al. (2024), when considering $W_1$ distance, the total error has the following form [8]

$$\frac{L_f L_{\text{score}}}{\sqrt{M}} + \sqrt{\delta},$$

where $M$ is the step number discussed in Section 3.2. To make the above term smaller than $\epsilon_{W_1}$, we require

$$M \geq \frac{L_f^2 L_{\text{score}}^2 d}{\epsilon_{W_1}^2} \text{ and } \delta \leq \epsilon_{W_1}^2.$$

As shown in (Chen et al., 2023d), we know that $\sigma_\delta^2 = \delta$ for VPSDE forward process. Furthermore, as shown in Lemma E.1, we need to choose $L_{\text{score}} = R^2/\sigma_\delta^4 = R^2/\delta^2$, which finally requires

$$M \geq \frac{L_f^2}{\epsilon_{W_1}^{10}}.$$

(b) Consistency Training Paradigm.

As shown in Remark 4.5 of Dou et al. (2024), they choose $\epsilon_{W_1} = n^{-1/d}$, where $n$ is the size of the training dataset and $M = n^{10/d}$. Hence, the discretization complexity of the CT paradigm is $M \geq 1/\epsilon_{W_1}^{10}$.

# E. Auxiliary Lemmas

In this section, we first prove some regularity results for $\nabla \log q_t(\cdot)$ under the VESDE setting. The following two results come from Yang et al. (2024) (Lemma E.1, E.2 and E.4) and are used to control $\mathbb{E}_{X_t \sim q_t}\left[\|\partial_t \nabla \log q_t(X_t)\|_2^2\right]$, which is useful in controlling the one-step error $W_2^2\left(qQ_{\text{ODE}}^{t_k', h_k'}, q\widehat{P}_{\text{ODE}}^{t_k', h_k'}\right)$.

**Lemma E.1.** *Assume Assumption 4.1. Then for any $t \in (0, T]$ and $X \in \mathbb{R}^d$ we have that*

$$\langle \nabla \log q_t(X), X \rangle \leq -\|X\|_2^2/\sigma_t^2 + R\|X\|_2/\sigma_t^2.$$

*In addition, we have*

$$\|\nabla \log q_t(X)\|^2 \leq 2\|X\|_2^2/\sigma_t^4 + 2R^2/\sigma_t^4,$$

*and*

$$\left\|\nabla^2 \log q_t(X)\right\|_2 \leq \left(1 + R^2\right)/\sigma_t^4.$$

**Lemma E.2.** *Assume Assumption 4.1. Then for any $t \in (0, T]$ and $X \in \mathbb{R}^d$ we have*

$$\|\partial_t \nabla \log q_t(X)\|_2 \leq \frac{g(t)^2}{\sigma_t^6} R^2(R + \|X\|_2).$$

**Lemma E.3.** *Suppose Assumption 4.1 holds. Let $\epsilon_{W_2} > 0$. (1) If considering VESDE with $\sigma_t^2 = t^2$, we choose the early stopping parameter $\delta = \frac{\epsilon_{W_2}}{\sqrt{d}}$, (2) If consider VPSDE, we choose $\delta = \epsilon_{W_2}^2/(\sqrt{d}(R \vee \sqrt{d}))$ then we have $W_2(q_\delta, q_0) \leq \epsilon_{W_2}$.*

**Proof.** For the VESDE forward process Equation (1), we know that $X_t := X_0 + \sigma_t z$, where $z \sim \text{normal}(0, I_d)$ is independent of $X_0$. Hence, for $\delta \lesssim 1$,

$$W_2^2(q_0, q_\delta) \leq \mathbb{E}\left[\|\sigma_\delta z\|_2^2\right] = \sigma_\delta^2 d.$$

Then, for the setting $\sigma_t^2 = t^2$, we can take $\delta \leq \frac{\epsilon_{W_2}}{\sqrt{d}}$.

For the VPSDE forward process, we directly use Lemma 20 of Chen et al. (2023d) to obtain the final results.

∎

---

[8]Here, we only discuss the dependence on $\epsilon_{W_1}$ and ignore $R, d$.

**Lemma E.4.** *Suppose that Assumption 4.1 hold. Let $(X_t)_{t \in [0,T]}$ denote the forward process Equation (1). Then, for all $t \geq 0$,*

$$\mathbb{E}\left[\|X_t\|_2^2\right] \leq d\sigma_t^2 + R^2$$

**Proof.** We know that $X_t = X_0 + \sigma_t Z$, where $Z \sim \mathcal{N}(0, I)$. Hence, we have

$$\mathbb{E}\left[\|X_t\|_2^2\right] = \mathbb{E}\left[\|X_0\|_2^2\right] + \sigma_t^2 d \leq d\sigma_t^2 + R^2.$$

$\blacksquare$

## F. Simulation Experiments and Simplified PFODE discussion

In this section, we support our Assumption 4.4 by using the simulation experiments. Since we do not have the closed-form consistency function for highly multi-modal target distribution, we simulate the consistency function $\boldsymbol{f^v}$ by running the following process starting from $\mathcal{N}(0, T^2 I_d)$ (We choose $T = 50$ in our experiments):

$$d\bar{Y}_{t'} = \frac{g(T - t')^2}{2} s_{\boldsymbol{\phi}}\left(\bar{Y}_{t'_k}, T - t'_k\right) dt', t' \in [t'_k, t'_{k+1}].$$

In this part, we use multi-modal Gaussian Mixture distribution as the target distribution. Since the ground-truth score function of Gaussian Mixture Distribution can be calculated, we directly use the ground-truth score function $\nabla \log q_t(\cdot)$ (where the target consistency function $\boldsymbol{f^v}$ is also determined with the ground-truth score function) to run the above process to approximate $\boldsymbol{f^v}$ (In Figure 1, $\boldsymbol{f}^{\text{ex}}$ is $\boldsymbol{f^v}$ in the main content).

For the Lipschitz constant of $\boldsymbol{f^v}$ at a fixed time $t'$, we calculate $\frac{\boldsymbol{f^v}(Y_1, t') - \boldsymbol{f^v}(Y_2, t')}{Y_1 - Y_2}$ as an approximation.

**Baseline.** Since we want to show that the Lipschitz constant of $\boldsymbol{f^v}(\cdot, t')$ has order $1/(T - t')$ at the beginning of the reverse process ($t'$ is small, $t$ is large), we provide a reference line (brown line in Figure 1) with $1/t$ Lipschitz constant. More specifically, after obtaining the Lipschitz constant $L_{f,0}$ of $\boldsymbol{f^v}(\cdot, t')|_{t'=0}$ (which is always the smallest one), we calculate the points at reverse time $t'$ (forward time $T - t'$) on the brown line by times $T/(T - t')$ on $L_{f,0}$.

**Observation and Discussion.** In this experiment, we use 3-modal and 4-modal Gaussian mixture distribution as the target distribution. As shown in Figure 1, the brown and the blue lines are almost the same at the beginning of the reverse process (the forward diffusion time $t$ is large). Hence, the simulation experiments support our Assumption 4.4.

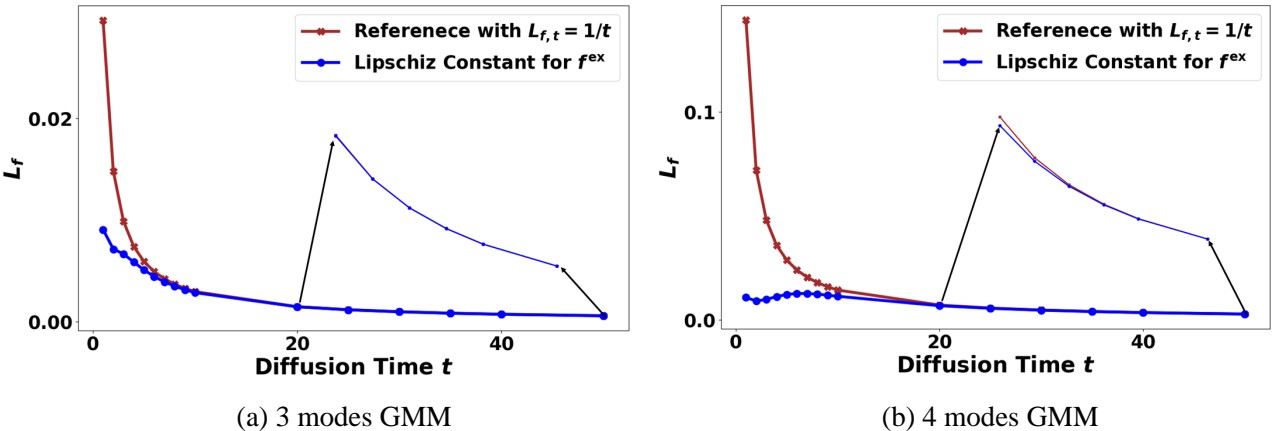

(a) 3 modes GMM  (b) 4 modes GMM

Figure 1: Simulation Experiments on the Lipschitz Constant.

