# OpenReview forum: "Improved Discretization Complexity Analysis of Consistency Models: Variance Exploding Forward Process and Decay Discretization Scheme"
_ICML.cc/2025/Conference — ICML 2025 poster_

### Official Review · Reviewer_Fjk9 · 2025-03-11

**Overall Recommendation:** 3

**Summary:**

The paper analyzed the consistency model of VE process and decay step size, and proved the discretization complexity of the consistency model.

**Claims And Evidence:**

The paper bridges the gap between theory and application of consistency models by analyzing the discretization complexity through mathematical derivation and support the main claims.

**Essential References Not Discussed:**

The author has thoroughly discussed the relevant literature.

**Experimental Designs Or Analyses:**

This paper does not include experiments. They provided a discretization complexity analysis of the consistency model in mathematics.

**Methods And Evaluation Criteria:**

This paper is a theoretical work and does not contain any empirical results.

**Other Comments Or Suggestions:**

I find no typos currently.

**Other Strengths And Weaknesses:**

**Strengths**
The authors also provide the 2-step Sampling analysis, which is widely used.

**Weaknesses**
The conclusion in the article is important for consistency distillation. As is well known, consistency training can independently train consistency models, which is missing in the article.

**Questions For Authors:**

Please see the weakness. Can the author clarify whether their conclusion can be extended to consistency training and continuous time consistency models.

**Relation To Broader Scientific Literature:**

In my opinion, this work is the first time it has closed the gap between the discretization complexity analysis for the consistency model and the practical setting. And it overcomes some limitations of prior work such as [1][2].

[1] Zehao Dou, Minshuo Chen, Mengdi Wang, and Zhuoran Yang. Theory of consistency diffusion models: Distribution estimation meets fast sampling. In Forty-first International Conference on Machine Learning, ICML 2024, Vienna, Austria, July 21-27, 2024.

[2] Junlong Lyu, Zhitang Chen, and Shoubo Feng. Sampling is as easy as keeping the consistency: convergence guarantee for consistency models. In Forty-first International Conference on Machine Learning, 2024.

**Theoretical Claims:**

As I am not an expert in diffusion model theory, it is difficult for me to keep up with some parts of the paper, so I did not check the correctness of all the theorems.

---

> ### Author Rebuttal · Authors · 2025-03-31
>
> Thank you for your valuable comments and suggestions. We provide our response to each question below.
>
> **Weakness & Suggestion: The analysis for consistency training and  continuous time consistency models.**
>
> As shown by the professional reviewer, the consistency training and continuous-time consistency models are both important part of of consistency models. In this part, we discuss some possible method to obtain the discretization complexity for this method.
>
> **Consistency Training.** If we can not obtain the pre-trained score function, we can construct an empirical score by using $n$ samples from the target data distribution $\\{X_{0,i}\\}\_{i=1}^n$
> $$
> s\_{\mathrm{emp}}(X_t ; t)=-\frac{1}{ \sigma_t^2}\left[X_t- \frac{\sum\_{i=1}^N \mathcal{N}\left(X_t ; X\_{0,i}, \sigma_t^2 I\right) X_{0,i}}{\sum\_{i=1}^N \mathcal{N}\left(X\_t ; X_{0,i}, \sigma_t^2 I\right)}\right],
> $$
> which has an explicit formulation, needs no additional training ([1] also use this formula) and converges to the ground-truth score function with a rate $n^{-1/d}$. Hence, we can replace the pretrained score function $s_{\phi}$ in eq. (4) with $s_{\mathrm{emp}}$. Then, the $\epsilon_{\text{score}}$ becomes $n^{-1/d}$ and achieve the guarantee for consistency models without a pre-trained score function under the VE process and EDM stepsize.
>
> We note that though this results does not rely on the pretrained score function, it use the reverse PFODE process of diffusion. On the contrary, the consistency training paradigm only use the forward diffusion process. Hence, the above result is not the discretization complexity of consistency training paradigm. To achieve this goal, one possible way is to use similar method with [1], which use $s\_{\mathrm{emp}}$ to construct a baseline consistency function (a bridge between the target data distribution and the consistency function learned by the consistency training paradigm)  instead of directly using it in the training objective function.  However, as shown in our Remark 4.10, the construction of the baseline consistency function run $M$-step PFODE instead of one-step PFODE (used in application), which leads a large discretization complexity $1/\epsilon_{W_1}^{10}$. Since this result is significant larger than our $1/\epsilon\_{W_2}^3$ results, we left the discretization complexity analysis (compareable with the CD paradigm) for the CT paradigm under the setting used in application.
>
> **Continuous-time consistency models.** Since continuous time model use $\frac{\mathrm{d} \boldsymbol{f}\_{\theta^{-}}(X\_t, t)}{\mathrm{d} t}$ instead of $\boldsymbol{f}\_{\theta^{-}}(X\_{t-\Delta t}, t-\Delta t))$ ($\Delta t$ is $h_{k+1}-h\_{k}$ in our work. Here we use the uniform stepsize for convenience), there are not well-defined discretization complexity $K=T/\Delta t$ for continuous-time models. However, due to the absence of $\Delta t$, the training process of continuous time models is less stable than discrete time consistency models, which is the core problem for continuous time models. Recently, [2] make a great effort to stabilize the training process of continuous time models.
>
> Thanks again for the comments on a broader area of consistency models and we will add the above discussion in our next version.
>
>
>
> [1] Dou, Zehao, Minshuo Chen, Mengdi Wang, and Zhuoran Yang. "Theory of consistency diffusion models: Distribution estimation meets fast sampling." In *Forty-first International Conference on Machine Learning*. 2024.
>
> [2] Lu, Cheng, and Yang Song. "Simplifying, stabilizing and scaling continuous-time consistency models." *arXiv preprint arXiv:2410.11081* (2024).

---

### Official Review · Reviewer_iimn · 2025-03-11

**Overall Recommendation:** 3

**Summary:**

The paper proposes a novel discretization complexity analysis of Consistency Models, by incorporating the variance exploding kernel and the non-uniform step size. The results are closer to diffusion models than previous methods, providing a better analysis of conistency models.

## Update after rebuttal
The several reviews and answers clarified some parts of the paper, and I will maintain my previous score.

**Claims And Evidence:**

The claims of achieving better complexity analysis are supported by the proofs. The framework represents more closely what is commonly done in the practice, which results in complexity results more close to the ones of diffusion models, which could help motivating the great empirical performance of consistency models.

**Essential References Not Discussed:**

N/A

**Experimental Designs Or Analyses:**

As the paper is mostly theoretical, there is no real experimental section, besides some simulations in Appendix F.

**Methods And Evaluation Criteria:**

Besides appendix F, there are no evaluations, as the claims are theoretical and supported by proofs.

**Other Comments Or Suggestions:**

Would be useful to name corollaries and lemmas in the same way between main text and appendix.

**Other Strengths And Weaknesses:**

Given the analysis from the paper, I wonder if one can derive practical considerations to design better consistency models. Having a discussion about this could make the paper more relevant for applied research.

**Questions For Authors:**

1- From your results, is it correct that the complexity decreases as $a$ approaches $\infty$? Would that mean that in practice, schedules with big $a$ should be preferred?

**Relation To Broader Scientific Literature:**

Consistency Models are a novel generative modeling framework which can achieve performance similar to diffusion models with significantly less sampling steps. Deepening our theoretical understanding of these models is relevant to further improve their performance.

**Theoretical Claims:**

The assumptions (4.1 to 4.4) are reasonable and in line with related literature. The results from theorem 4.7, corollaries 4.8 and 4.12 with proofs in appendix B seem correct.

---

> ### Author Rebuttal · Authors · 2025-03-31
>
> Thank you for your valuable comments and suggestions. We provide our response to each question below.
>
> **Weakness 1: The guidance on the design of better consistency models.**
>
> This paper makes the first step to elucidate the design space of consistency models under the different diffusion processes and reveals their different advantages and disadvantages, which will heavily influence the discretization complexity and is fundamental in designing the consistency models. For the VP-based consistency models, the early stopping parameter $\delta$ has order $\epsilon_{W_2}^2$, which is worse than the VE-based consistency models ($\delta$ has order $\epsilon_{W_2}$) and is the source of large discretization complexity. However, the VE-based consistency models also have their disadvantage: the polynomial diffusion time $T$, which is much larger than $T=\log(1/\epsilon)$ for the VP-based consistency models and introduces additional $\epsilon_{W_2}$ dependence.
>
> Hence, from the discretization perspective, a better consistency model should enjoy a Logarithmic $T$ and a $\delta$ with order $\epsilon_{W_2}$, which would lead to better complexity results. We note that the rectified flow-based one-step models have this potential. We will add the above discussion to our next version and view the design of better consistency models as important future work.
>
> **Question 1: The choice of $a$.**
>
> Our results show that with a larger $a$, the discretization complexity is better than the uniform discretization scheme ($a=1$).This phenomenon is also observed in the empirical work EDM [2], which observed that when $1\leq a\leq 7$, a larger $a$ will help the diffusion models to achieve better performance (Figure 13 (c) of [2]). When $a$ is larger than $7$, the improvement is not significant. Consistency models follow the choice of $a$ in EDM. In our Theorem 4.7, we also show that with $a=7$, the discretization complexity  has order $1/\epsilon_{W_2}^{23/7}$, which is close to the $1/\epsilon_{W_2}^{3}$ of exponential decay stepsize.
>
> [1] Lyu, Junlong, Zhitang Chen, and Shoubo Feng. "Sampling is as easy as keeping the consistency: convergence guarantee for Consistency Models." In *Forty-first International Conference on Machine Learning*. 2024.
>
> [2] Karras, Tero, Miika Aittala, Timo Aila, and Samuli Laine. "Elucidating the design space of diffusion-based generative models." *Advances in neural information processing systems* 35 (2022): 26565-26577.

---

### Official Review · Reviewer_a46D · 2025-03-14

**Overall Recommendation:** 1

**Summary:**

This paper examines the convergence of the consistency model under the VE process with a decaying step size. It focuses on consistency distillation and establishes convergence results based on the Wasserstein distance between the generated and target distributions. Additionally, it demonstrates that 2-step sampling enhances discretization efficiency.

**Claims And Evidence:**

The main result, Theorem 4.7, heavily depends on Assumption 4.4, which lacks supporting evidence. See below for details.

**Essential References Not Discussed:**

No.

**Experimental Designs Or Analyses:**

In Appendix F, the calculation of the Lipschitz constant is not clearly explained.

**Methods And Evaluation Criteria:**

Theory paper, not applicable.

**Other Comments Or Suggestions:**

The author could analyze the Lipschitz constant of the consistency function at $ (x,t) = (0,T) $ for the bimodal Gaussian mixture model $ 0.5 N(-1,\sigma^2) + 0.5 N(1,\sigma^2) $ for a small $ \sigma $.

**Other Strengths And Weaknesses:**

This paper examines the Lipschitz coefficient of the consistency function, a crucial step in understanding consistency models.

**Questions For Authors:**

None

**Relation To Broader Scientific Literature:**

This paper investigates the convergence of the consistency model under the variance-exploding process, whereas prior work primarily focuses on the variance-preserving process.

**Theoretical Claims:**

I find Theorem 4.7 to be not very informative. According to Appendix B, the first term in the error decomposition is $ L_{f,0} R $, which does not asymptotically converge to zero. Therefore, the condition on $ L_{f,0} $ must be strict: even if $ L_{f,0} = O(1) $, the error bound in Theorem 4.7 remains $ O(R) $. Since $ R $ represents the diameter of the target distribution’s support, an error bound of $ O(R) $ is trivial. This paper only establishes $ L_{f,0} = R/T $ for the Gaussian distribution, which is quite limited. To derive a more meaningful result, the paper should rigorously demonstrate that $ L_{f,0} = R/T $ holds for a broader class of distributions rather than merely assuming it (Assumption 4.4).

---

> ### Author Rebuttal · Authors · 2025-03-31
>
> Thank you for your valuable comments and suggestions. We provide our response to each question below.
>
> **Q1: Theoretical Claims: The discussion on $L_{f,0}$ assumption and remove it.**
>
> Following the suggestion of the professional reviewer, we consider the $L_{f,0}$ in 2-mode GMM in the following **Suggestion 1** and show $L\_{f,0}$ has the order $1/T$, which is necessary for a $W\_2$ guarantee. In this part,  we mainly discuss how to remove this assumption. We prove that if considering  a weaker $W\_1$ guarantee,  **we can remove $L\_{f,0}=O(R/T)$ assumption and achieve a  $L_f^{1+1/a}/\epsilon\_{W_1}$ result**:
>
> When considering $W_1$ guarantee,  the first term of line 615 (appendix B) becomes $L_fW\_1(\mathcal{N}(0,T^2I_d),q_T)$ (Using uniform $L_f$ instead of $R/T$). Different from the $W_2$ distance, the $W_1$ distance can be bounded by weight TV distance (Case 6.16 [1]):
> $$
> W_1(\mathcal{N}(0,T^2I_d),q_T)\leq R\mathrm{TV}(\mathcal{N}(0,T^2I_d),q_T)\leq R^2/T,
> $$
> where the second inequality follows the fact of [2]. Hence, we do not require $L_{f,0}=O(R/T)$. The other proof is exactly the same with $W_2$ distance. To guarantee $L_fW\_1(\mathcal{N}(0,T^2I_d),q_T)$ smaller than $\epsilon_{W_1}$,  we require $T\ge L_fR^2/\epsilon\_{W_1}$, which is the source of additional $L_f^{1/a}$. We will add the above result in the next version.
>
> **Q2: Experimental Analysis: the calculation of $L_{f,0}$ in simulation experiments.**
>
> Since $L_{f,0}$ can be obtained by calculate the F-norm of $\nabla_{Y_0}\boldsymbol{f}(Y_0,0)$, we calculate the following equal to approximate it
> $$
> \left|\frac{\boldsymbol{f}^{\boldsymbol{v}}\left(Y_{t^{\prime}}, t^{\prime}\right)-\boldsymbol{f}^{\boldsymbol{v}}\left(Y_{t^{\prime}}+\Delta Y, t^{\prime}\right)}{\Delta Y}\right|,
> $$
> where $Y_{t'}\sim q\_{T-t'}$ (sample $1000$ times and take average) and $\Delta Y = 0.01$.
>
> **Suggestion 1: The Lipschitz constant (Mixture of Gaussian).**
>
> We sincerely thanks again for the comments. We consider 2-mode GMM $X_0 \sim 1/2N(\mu, \sigma^2I_d)+1/2N(-\mu, \sigma^2I_d)$. The score has the following form (Appendix A.2 of [3], we transform it from VP to VE)
> $$
> \nabla \log q_t(X_t)=\tanh(\frac{\mu^{\top} X_t}{\sigma_t^2+\sigma^2}) \frac{\mu}{\sigma_t^2+\sigma^2}-\frac{X_t}{\sigma_t^2+\sigma^2}.
> $$
> Since $f^{\mathrm{ex}}(Y_0,0)$ the associate backward mapping of the following PFODE (in the following part, we ignore the superscript of $t'$):
> $$
> dY_t=\tanh(\frac{\mu^{\top} Y_t}{(T-t)^2+\sigma^2}) \frac{\mu(T-t)}{(T-t)^2+\sigma^2}-\frac{Y_t(T-t)}{(T-t)^2+\sigma^2}dt,
> $$
> we need to solve it to obtain $f^{\mathrm{ex}}(Y_0,0)$. Since the score is highly nonlinear, it is hard to obtain a closed-form solution. There are two choices to overcome this hardness.  The first choice is to do simulation experiments to simulate the solution (Appendix F).
>
> The second choice is to add some assumptions on the target data to simplify the above ODE. We assume $\mu$ is smaller enough to guarantee $\tanh \left(\frac{\mu^{\top} Y_t}{(T-t)^2+\sigma^2}\right)$ can be approximated by $\frac{\mu^{\top} Y_t}{(T-t)^2+\sigma^2}$, which simplify PFODE to a linear ODE  (in fact, the distribution gradually closes to Gaussian)
> $$
> \mathrm{d} Y_t=\left(\frac{\mu^{\top} \mu Y_t(T-t)}{\left((T-t)^2+\sigma^2\right)^2}-\frac{Y_t(T-t)}{(T-t)^2+\sigma^2}\right) \mathrm{d} t,
> $$
> which have the following solution
> $$
> Y_t=Y_0\underbrace{\left(\sqrt{\frac{\sigma^2+(T-t)^2}{\sigma^2+T^2}} \cdot \exp \left(\frac{\mu^2}{2}\left(\frac{1}{\sigma^2+(T-t)^2}-\frac{1}{\sigma^2+T^2}\right)\right)\right)}\_{C(t)}.
> $$
> The above results indicate
> $$
> Y_T=Y_0\left(\sqrt{\frac{\sigma^2}{\sigma^2+T^2}} \cdot \exp \left(\frac{\mu^2}{2}\left(\frac{1}{\sigma^2}-\frac{1}{\sigma^2+T^2}\right)\right)\right)
> $$
> Taking the derivative of $Y_0$, we know that the $L_{f, 0}$ have order $1 / T$. This result also matches our intuition that $Y_0$ has a large variance (order $T^2$ ), and we need to multiply a $1 / T$ to avoid the influence of large variance (lines 282-287).
>
> **Further Discussion on the error of linear approximation.**  The above part makes a linear approximation to simplify the ODE when $\mu$ is close to $0$, which introduces some small errors. Assuming $Y_0\sim q_T= 1/2N(\mu, (T^2+\sigma^2)I)+1/2N(-\mu, (T^2+\sigma^2))$. For the variance, $Y_T=Y_0C(T)$ recover $\sigma^2$. For the mean, the recover $\mu$ of the above consistency function is approximately  $\mu\sqrt{\sigma^2/(\sigma^2+T^2)}$, which is smaller than $\mu$. However, since assuming $\mu$ is close to $0$, this error term is small and is possibly introduced by the nonlinear term.
>
> We will add the above discussion in the next version.
>
> [1]  Villani, Cédric. *Optimal transport: old and new*. Vol. 338. Berlin: springer, 2008.
>
> [2] Yang et al,. Leveraging Drift to Improve Sample Complexity of Variance Exploding Diffusion Models. NeurIPS 2024.
>
> [3] Shah et al,. Learning mixtures of gaussians using the DDPM objective. NeurIPS 2023.

---

> > ### Comment · Reviewer_a46D · 2025-04-03
> >
> > Thank you for your response. Please see my comments below:
> >
> > 1. **Discussion on the Lipschitz condition**:
> >    I find both inequalities in question to be problematic.
> >    - The **first inequality** relies on the fact that $W_1(P_1, P_2) \le R \cdot TV(P_1, P_2)$, where $R$ is the diameter of the support of both distributions. However, in your setting, both $N(0, T^2 I_d)$ and $P_T$ are clearly unbounded, making this inequality inapplicable.
> >    - The **second inequality** uses convergence results from a paper that operates under a different setup. Specifically, [2] analyzes a forward SDE with a **drift term**, while the forward SDE in your paper does **not** include a drift. Therefore, the results from [2] do not apply here.
> >
> > 2. **Empirical evaluation of the Lipschitz constant and Assumption 4.4**:
> >    Assumption 4.4 posits that $\sup_y ||\nabla_y f(y, 0)|| \le L_{f,0}$. However, according to the rebuttal, the experimental evaluation computes $E_{y \sim p_T}[||\nabla_y f(y, 0)||]$. These are not equivalent; in fact, $\sup_y ||\nabla_y f(y, 0)|| \ge E_{y \sim p_T}[||\nabla_y f(y, 0)||]$. So, the empirical evaluation does not support the assumption.
> >
> > 3. **The 2-mode GMM example**:
> >    - **Simulation of the Lipschitz constant**: As noted above, there is a mismatch between the theoretical assumptions and the empirical estimation of the Lipschitz constant.
> >    - **Error from linear approximation**: I have several concerns here:
> >      1. Grönwall’s inequality suggests that the approximation errors can have **exponential effects** on the solution. This raises doubts about the validity of a linear approximation.
> >      2. It is unclear whether the term $\frac{\mu^\top Y_t}{(T - t)^2 + \sigma^2}$ can be treated as small, even if $\mu$ is small:
> >         - $Y_t$ may be unbounded;
> >         - $(T - t)^2 \to 0$ as $t \to T$;
> >         - $\sigma$ could also be small.
> >      3. Even if we accept the linear approximation, the resulting Lipschitz constant **grows exponentially** as $\sigma \to 0$, leading to a vacuous theoretical bound.
> >
> > Given these issues, I will maintain my evaluation.

---

> > > ### Author Response · Authors · 2025-04-03
> > >
> > > We sincerely thank the professional and helpful reviewer for further feedback and comments. We provide our response to each question below.
> > >
> > > **Q1: The discussion of $W_1$ results.**
> > >
> > > As pointed out by the helpful reviewer, the distribution $\mathcal{N}(0,T^2I)$ and $q_T$ is unbounded. Then, we can not remove this second part of  Assumption 4.4 ($R/T$ assumption) even considering the $W_1$ distance (Hence, we will not add this discussion in our paper. Thanks again!). To verification our assumption, we do more simulation experiments with the uniform sampling $Y$ (instead of sampling $Y_t$ according to $q_{T-t'}$) to simulate the  $\sup\_y\left\\|\nabla_y f(y, 0)\right\\|$ instead of $E\_{y \sim q\_T}\left\[\left\\|\nabla_y f(y, 0)\right\\|\right\]$ and show that in a large range of $Y$, the $L_{f,0}$ have the order $1/T$ ($Y\in \\{1,2,3,...,40\\}$).
> > >
> > > **Q2: The further simulation experiments using uniform sampling instead of sampling according to $q_t$.**
> > >
> > > As mentioned in **Q1**, in this part, we do simulation experiments on  3 GMM with different $Y$ (and $\Delta Y=0.01$) to verify the Lipschitz constant has order $1/T$ in a large range of $Y$ ($Y\in \{1,2,3,...,40\}$). The kindly reviewer can see the simulation experiments in the following link.
> > >
> > > Simulation Experiment Link: https://anonymous.4open.science/r/ICML_Consistency_Simulation-8AF6/Rebuttal_Simulation_Consistency.pdf
> > >
> > > **Q3: The linear approximation.**
> > >
> > > (a) Since the closed-form solution for the PFODE with a nonlinear score function is hard to obtain, we make a linear approximation in the nonlinear score of 2-GMM to clearly discuss the order of Lipschitz constant (This linear approximation has been used in previous theoretical works on diffusion models with GMM distribution due to the difficult nonlinear terms (Lemma 8 of [1])). As discussed by the reviewer, the linear approximation will introduce some approximation errors. For this error, at the end of our rebuttal, we show that the obtained consistency function ($C(T)$) can approximately recover the target 2-GMM.
> > >
> > > (b) The influence of $\sigma$.
> > >
> > > For the variance term of GMM, since the current image datasets is usually normalized, $\sigma^2$ is not close to $0$ in application (then, we can view it as a constant, such as $1$). Hence, it will not introduce an additional exponential term.
> > >
> > > (c) The choice of $\mu$.
> > >
> > > We know that, with a high probability, $Y_t$ falls in the range $[-3(\sqrt{(T-t)^2+\sigma^2}), 3(\sqrt{(T-t)^2+\sigma^2})]$  (since $\mu$ is close to $0$, the 2-GMM is close to Gaussian)(We also defined a truncated operator in this interval for $Y_t$). Then, we choose a small enough $\mu$ that guarantees $\mu^\top Y_t$ with the truncated $Y_t$ is smaller (for $Y_t$ out of this interval, intuitively, it can control by the tail bound of Gaussian and introduce additional truncated error). Hence, the linear approximation is possible and will not introduce an exponential term (with a constant $\sigma$ in (b)).
> > >
> > > The nonlinear score is hard to deal with in the area of diffusion models, and we sincerely hope the above discussion can address the concerns of the professional reviewer. We also hope that the insightful reviewer will re-evaluate this work based on our discussion.
> > >
> > > Best,
> > >
> > > Authors
> > >
> > >  [1] Shah et al,. Learning mixtures of gaussians using the DDPM objective. NeurIPS 2023.

---

### Official Review · Reviewer_KKHE · 2025-03-24

**Overall Recommendation:** 4

**Summary:**

This paper aims to provide a theoretical explanation for the strong empirical performance of consistency models — specifically focusing on how many discretization steps $K$ are needed during training to guarantee high-quality one-step sampling at test time. Prior theoretical analyses of consistency models typically used variance-preserving (VP) forward processes with uniform steps, leading to large and possibly unrealistic complexity bounds. This work, instead, targets the variance-exploding (VE) forward process and the EDM (decay) time-step scheduling. Under these more practical assumptions (matching real applications in, e.g., Karras et al., 2022 or Song et al., 2023), the authors derive improved discretization complexity bounds – polynomial in $O(1/\varepsilon)$ with exponents significantly better than previous results. They also show that 2-step sampling (a widely used trick in consistency models) can further reduce the required number of steps to achieve a given Wasserstein-2 error.

**Claims And Evidence:**

In this paper, the authors claimed that analyzing VESDE plus EDM steps yields a polynomial discretization bound for consistency models that is significantly smaller than in previous theoretical studies, and this complexity is close to that of the best known diffusion results. In addition, 2-step sampling further reduces the exponent in $\varepsilon$.

To show these claims, the authors provided rigorous proof in the main text and appendix. They compared the final complexity expressions to older results, showing strict improvement. Besides, simulation experiments for multi-modal Gaussian distributions illustrate that their key assumption on Lipschitz constants is possible.

**Essential References Not Discussed:**

Nothing crucial seems missing. The standard relevant theoretical diffusion or consistency references appear.

**Ethical Review Concerns:**

No ethical concerns since it is a completely theoretical work.

**Experimental Designs Or Analyses:**

There are no experiments needed in this theoretical paper.

**Methods And Evaluation Criteria:**

As a theoretical paper, there is no benchmark or datasets needed.

**Other Comments Or Suggestions:**

Please refer to the "Weaknesses" section.

**Other Strengths And Weaknesses:**

Strengths:
The focus on VE and EDM steps is precisely the realistic setting used in modern SOTA consistency models, bridging earlier theoretical-limitation criticisms. It is a good progress to achieve $\tilde{O}(\frac{1}{\varepsilon^{3+2/a}})$ with 2-step sampling compared with the previous $O(1/\varepsilon^7)$. The paper is well-organized, with main results clearly stated and strictly proved.

Weaknesses:
1. The entire analysis relies on an assumption that the score approximation is sufficiently accurate. Is it possible for the authors to remove this assumption and handle the end-to-end training complexity?
2. The multi-step analysis is restricted to 2 steps; though that’s the main empirical scenario, I still want to ask about the scenario with more steps or sampling schedules. Have you considered a more general $N$-step approach for consistency? Could that yield further improvements or do you expect diminishing returns after 2 steps?
3. Could your “time-dependent lemma” approach be extended to other step-size patterns beyond EDM (like a piecewise approach)? Are there potential further gains if we do a more sophisticated scheduling than a single exponent $a$.

**Questions For Authors:**

Please refer to the "Weaknesses" section.

**Relation To Broader Scientific Literature:**

This is the first analysis that specifically uses VE forward SDE plus a decaying step approach for consistency. It corrects prior mismatches in theoretical assumptions vs. real usage. The authors connect the final complexity to that of diffusion, bridging a gap that older works left open. They cite relevant works on diffusion complexity (Song et al., Gao & Zhu, Chen et al.), on prior consistency theory (Dou et al. 2024, Li et al. 2024, Lyu et al. 2024). They also mention Karras et al. for EDM steps. Therefore, I would like to say the references are quite comprehensive.

**Theoretical Claims:**

The paper states each assumption explicitly, references prior standard assumptions (like bounded support for data or Lipschitz continuity of the consistency function). The proofs revolve around standard SDE manipulations, approximate PDE expansions, and the idea that “time-dependent” bounding of the score drift is more precise.

---

> ### Author Rebuttal · Authors · 2025-03-31
>
> Thank you for your valuable comments and suggestions. We provide our response to each question below.
>
> **Weakness 1: The approximated score and consistency function error (end-to-end analysis).**
>
> In this work, we assume the pretrained score and consistency function are accurate enough to achieve the final discretization complexity. Though they are standard assumptions in the complexity analysis area, as the friendly reviewer mentioned, the end-to-end analysis is also important, and we can use the current estimation error analysis results to achieve this goal. More specifically, for the approximated score, we use the results of [1] and replace $\epsilon_{score}$ with $n\_{score}^{-2/d}$ (where $n\_{score}$ is the number of data used to train the score function). For the approximated score function, we use the result of [2] and replace $\epsilon\_{cm}$ with $n\_{cm}^{-1/2(d+5)}$. Then, we can obtain the end-to-end complexity analysis.
>
> **Weakness 2: The Results of Multi-step Sampling Algorithm.**
>
> In fact, our analysis can be extended to $N$-step sampling algorithm and can achieve nearly $L_{f}/\epsilon\_{W_2}^{3+1/a}$ (which is better than Thm. 4.7 and Coro 4.12) under the EDM stepsize. We use $3$-step sampling algorithm as an example ($\tau_1=T,\tau_2=3T/4, \tau\_3=T/2$). Under this setting, the result becomes (here we ignore $\epsilon\_{score}$, $\epsilon_{cm}$, $R,d $ and focus on the dominated term)
> $$
> \delta+1/T^3++L_f (T / \delta)^{\frac{1}{a}} /\left(K \delta^2\right).
> $$
> To guarantee the above term smaller than $\epsilon_{W_2}$, we require $\delta=\epsilon\_{W_2}$ and $K\ge \frac{L_fT^{1/a}}{\delta^{2+1/a}\epsilon\_{W_2}}=\frac{L_fT^{1/a}}{\epsilon\_{W_2}^{3+1/a}}$, which is the same with the one-step and two-step sampling algorithms. However, 3-step algorithm only require $T\ge 1/\epsilon\_{W_2}^{1/3}$, which is better than $1/\epsilon\_{W_2}$ of 1-step and $1/\epsilon\_{W_2}^{1/2}$ of 2-step. Hence, the discretization complexity for $3$-step sampling algorithm is $L_f/\epsilon\_{W_2}^{3+4/(3a)}$, which is better than $2$-step algorithm. The above steps can be extended to $N$ steps, and the influence of $T$ decreases, and finally, $T$ does not affect the discretization complexity, which leads to a $L_{f}/\epsilon\_{W_2}^{3+1/a}$ results. We will add the above discussion in our next version.
>
> **Weakness 3: The Discretization Complexity for Piecewise Discretization Scheme (beyond the EDM with single $a$).**
>
> Before providing the complexity result for the piecewise discretization scheme, we first discuss the performance of the EDM scheme in the application (Consistency models follow the choice of $a$ of EDM.). EDM [3] shows that when $1\leq  a\leq 7$, a larger $a$ will help the diffusion models to achieve better performance (Figure 13 (c) of [2]), which also matches our theoretical results. However, when $a$ is larger than $7$, the improvement is not significant and even becomes worse [3]. As a result, the exponential decay stepsize is theoretically friendly and is not widely used in applications. One possible explanation is that at the end of the reverse process, diffusion models generate image details and require a small stepsize (However, the exponential decay stepsize is too large.)
>
> Hence, we can design a two-stage discretization scheme: (a) when $t'\in [0, T-1]$, we use the exponential decay stepsize; (b) when $t'\in (T-1, T-\delta]$, we use the EDM stepsize. With this scheme, the discretization complexity becomes  $L_{f}/\epsilon\_{W_2}^{3+1/a}$, which is better than Thm. 4.7 with EDM (single $a$). This result shows the improvement of the two-stage discretization scheme from the theoretical perspective, and we leave the empirical application of this scheme as an interesting future work. We will add the above discussion in our next version.
>
> [1]Oko, Kazusato, Shunta Akiyama, and Taiji Suzuki. "Diffusion models are minimax optimal distribution estimators." In *International Conference on Machine Learning*, pp. 26517-26582. PMLR, 2023.
>
> [2]Dou, Zehao, Minshuo Chen, Mengdi Wang, and Zhuoran Yang. "Theory of consistency diffusion models: Distribution estimation meets fast sampling." In *Forty-first International Conference on Machine Learning*. 2024.
>
> [3]Karras, Tero, Miika Aittala, Timo Aila, and Samuli Laine. "Elucidating the design space of diffusion-based generative models." *Advances in neural information processing systems* 35 (2022): 26565-26577.

---

### Decision · Program_Chairs · 2025-05-01

**Decision:**

Accept (poster)

**Comment:**

This paper studies discretization complexity of consistency diffusion models. Compared to the existing results, the settings in the paper closely aligns with the practice: 1) The forward process is a variance exploding SDE and 2) the time discretization follows EDM. Under assumptions, the established complexity bounds improve over existing ones.

Theoretical contributions are well recognized by reviewers, including the problem setup to reflect real applications, novel analysis to address the challenges, and clear presentation of the main results. These strength indeed justifies an acceptance of the paper.

However, there are outstanding weaknesses preventing a firm acceptance. The concern centers around the assumptions and the scope of Theorem 4.7. In particular, the authors assume that the clean data distribution is compactly supported and the consistency function $f$ is Lipschitz continuous in $Y$ at $t = 0$. These two assumptions on their own do not directly contradict. Nonetheless, later examples (example 4.5 and the simulation in Appendix F) violate the compact data assumption to argue the Lipschitz continuity. This leaves the study much weaker, even though the theoretical results remain valid. Additional simulation results do not help to reinforce the theoretical part.

In order to resolve the issues, my suggestion is two-fold: 1) relax Assumption 4.1 to light-tailed distributions. In that case, we can use a truncation argument to restrict our attention to a bounded domain. Then the discussion on Gaussian and Gaussian mixture models are supported under the assumption, or 2) remove the discussion on Gaussian and Gaussian mixture models, but try to show the clean data regularity leads to Assumption 4.5 (at least, provide a reasonable example, say the clean data distribution is Holder continuous on the compact domain). That being said, either of the fix requires an another round of review. Therefore, I am recommending a weak acceptance.